# Nanoscale chemical reaction exploration with a quantum magnifying glass

Katja-Sophia Csizi[1,3], Miguel Steiner [1,2,3] & Markus Reiher [1,2] ✉

Nanoscopic systems exhibit diverse molecular substructures by which they facilitate specific functions. Theoretical models of them, which aim at describing, understanding, and predicting these capabilities, are difficult to build. Viable quantum-classical hybrid models come with specific challenges regarding atomistic structure construction and quantum region selection. Moreover, if their dynamics are mapped onto a state-to-state mechanism such as a chemical reaction network, its exhaustive exploration will be impossible due to the combinatorial explosion of the reaction space. Here, we introduce a "quantum magnifying glass" that allows one to interactively manipulate nanoscale structures at the quantum level. The quantum magnifying glass seamlessly combines autonomous model parametrization, ultra-fast quantum mechanical calculations, and automated reaction exploration. It represents an approach to investigate complex reaction sequences in a physically consistent manner with unprecedented effortlessness in real time. We demonstrate these features for reactions in bio-macromolecules and metal-organic frameworks, diverse systems that highlight general applicability.

Molecular structures at the nanoscale exhibit versatile functions that allow them to fulfill complex tasks such as mechanical work, specific chemical reactions, electronic processing, light harvesting. Nanoscopic systems can also act as highly efficient catalysts; examples are found in (metallo)enzymes[1,2], (metal)-organic frameworks (MOFs)[3–5], and encapsulated catalysts[6,7]. A detailed understanding of chemical processes in nanoscopic systems by elucidation of their reaction mechanisms at an atomistic level is of fundamental importance[8,9], especially in the context of catalyst design. However, reaction mechanism elucidation in such systems is a very challenging task due to the complexity induced by system size. This complexity manifests itself in two ways: First, interaction models (typically, from electronic structure theory) are required that can cope with the sheer molecular size and, at the same time, describe the breaking and forming of chemical bonds accurately. Second, the many (nuclear) degrees of freedom span a configuration space that prohibits exhaustive conformer and reaction-path searches.

Regarding the first challenge, quantum mechanical/molecular mechanical (QM/MM) hybrid models[10–14] represent the only viable approach towards routine modeling of non-periodic heterogeneous systems. In QM/MM models, the computationally expensive QM description is confined to a comparatively small reaction center, whereas its (large) environment is described by a classical interaction potential such as a molecular mechanics force field[15]. The neglect of the electronic degrees of freedom for the major part of the composite system is often a valid approach, because the breaking and forming of chemical bonds, which entail significant electronic and structural rearrangements, are local phenomena. Machine learning potentials (MLPs) (see, for instance, Ref. 16 and references therein) have been considered for replacing the QM approach in these applications, because they can be evaluated almost as fast as classical force fields, but they can deliver energies with a reliability comparable to that of the QM reference[17,18].

Whereas QM/MM models are well known to be valuable for the study of reactions in well-studied systems such as proteins and nucleic acids[11,19–21] or MOFs[22,23], their transferability to general nanoscopic structures (examples are surface catalysts or other biomacromolecules such as polyhydroxybutyrates) remains limited. This is mainly due to

[1]ETH Zurich, Department of Chemistry and Applied Biosciences, Vladimir-Prelog-Weg 2, 8093 Zurich, Switzerland. [2]ETH Zurich, NCCR Catalysis, Vladimir-Prelog-Weg 2, 8093 Zurich, Switzerland. [3]These authors contributed equally: Katja-Sophia Csizi, Miguel Steiner. ✉e-mail: mreiher@ethz.ch

the lack of suitable classical force field parametrizations for general system classes (or due to a lack of data needed for (re)parametrization of MLPs). Generalist interaction models aim at alleviating this lack of parametrizations[24–29], but may suffer from the general problem that a reduced-dimensional model cannot achieve arbitrarily high accuracy for arbitrary molecules[30]. We have addressed this shortcoming by introducing system-specific parametrizations such as the system-focused atomistic models (SFAM) with force fields[31] and lifelong MLPs[32]. Both types of fast interaction models can be constructed automatically for any chemical species as their parameters are determined by reference to generated first-principles data. We have extended SFAM towards a general QM/MM framework (QM/SFAM)[33] with an automated pipeline for atomistic structure preparation[34] and QM region selection[33]. Automation is the key ingredient to ensure reproducibility and easy usability for routine application to chemically diverse nanoscopic systems.

Even with fast, system-focused, automatically generated QM/MM hybrid models, sampling of molecular structures on a high-dimensional potential energy surface (PES) remains to be a challenge, which can impede reaction mechanism elucidation and chemical discovery. A truncation of the search space not only introduces crucial limitations and severe biases, it may require some prior knowledge—either from experiment or from the existing literature. Hence, this becomes a time-consuming task, usually only achievable by domain experts. Enhanced sampling approaches applied to reactions such as metadynamics[35], boxed molecular dynamics[36], coordinate-driving-molecular-dynamics[37], or the nanoreactor[38] overcome these problems only partially as they can suffer from method-inherent biases such as the choice of collective coordinates or of initial conditions. By contrast, immediate human insight would be available if the sampling process had been presented to an operator in a way that is easy to grasp and that facilitates interactive manipulation. Here, we provide a solution that immerses an operator into a complex simulation task at the nanoscale so that human intervention can enhance chemical process elucidation. In these cases, the interaction of an operator with the exploration and its manipulation is an explicitly intended bias, which can be complemented with unbiased algorithms for confirmation of the original hypothesis.

We advocate for a solution that combines fast automated QM/MM hybrid model construction with (i) an operator-set focus on an arbitrary structural region within a nanoscopic structure and (ii) autonomous quantum chemical exploratory calculations for the resulting QM region. As a result, we obtain a magnifying glass through which one can investigate an electronic quantum subsystem embedded into a classical framework. Specifically, the focus of the magnifying glass may be put on any part of a nanoscopic system due to full automation of all technical procedures. Regarding the second challenge mentioned above, a combination with autonomous reaction network exploration[39–45] is then necessary to address the huge reaction space, as it allows one to consider orders of magnitude more structures than accessible by manual inspection. For example, we have developed mechanism exploration strategies[46–49] for the systematic, open-ended exploration of chemical reactions in an unbiased way solely based on the first principles of quantum mechanics.

In general, automated chemical reaction network (CRN) explorations are based on algorithms that locate transition states of all possible reactions for a given set of reactants[35,38,46–48,50–56]. However, these approaches are computationally demanding already for reaction mechanisms of medium-sized organic[57] and inorganic[58] compounds, and for small transition metal complexes[59], requiring years of computing time[43] or specialized algorithms applicable to nanoscopic systems (as, for instance, the microiterative anharmonic downward distortion following approach introduced in Ref. 60, and the artificial-force-induced-reaction ansatz based on ONIOM[61] introduced in Ref. 62). Hence, the aforementioned second challenge, namely the

explosion of configuration space with system size, requires another ansatz to cope with the size of nanoscopic systems. Real-time manual quantum chemical explorations[63,64] (based on ultra-fast semi-empirical quantum chemical calculations[65–67] executed within milliseconds) can, in principle, be a solution to this problem, possibly augmented with haptic feedback[65,68–70] and/or virtual reality[71–73]. A continuous instant-feedback loop of calculated properties (such as gradients (forces) on nuclei, atomic charges, energies) facilitates immersion into the exploratory process. As a result, reactive channels can be efficiently explored and subjected to further optimization and refinement. However, real-time quantum chemical calculations have been limited so far to molecular systems of up to about 150 atoms, because these computations must finish at very high frequency in order to be experienced by an operator in real time (note that this term must not be mistaken for how it is used in real (compared to imaginary) time molecular dynamics simulations). The definition of 'real time' in our context of interactive quantum mechanics depends on how we perceive data as human beings: it requires an update rate for quantum chemical results of about 25 Hz for visual feedback and about 1 kHz for haptic feedback (since our tactile sense is more sensitive than the human eye) in order to achieve real-time feedback for a smooth experience. For larger systems, this could be realized only with purely classical (force field) models[74–77] so far. However, resorting to purely classical models significantly limits the application range of interactive systems, as they can no longer adjust to any arbitrary reactivity situation (which is the unique advantage of a model based on the quantum mechanical description of electrons in the external field of the atomic nuclei).

In this work, we introduce a framework, the Quantum Magnifying Glass (QMG), which combines the automated construction of system-focused QM/MM hybrid models with automated reactivity exploration algorithms and real-time quantum chemistry to address the dimensionality problem associated with studying chemical reactions in complex molecular environments. As the unbiased sampling of the PES of a nanoscopic is computationally unfeasible, we introduce an intended bias focused on the given problem. Within the QMG, this bias can take the form of (i) interactive manipulations of structures in real-time by an operator and (ii) a priori defined reaction heuristics that guide an automated exploration. The former comes with the advantage of offering instant feedback on the reactivity of a system, allowing for an adjustment and hence improvement of the exploration coverage. The latter can be advantageous because introducing reaction rules can guide an initial automated mechanism exploration that subsequently extends its scope by systematically lifting or altering these reaction rules. We note that such biases are also present in seemingly unbiased enhanced sampling techniques (consider, for instance, the choice of velocities for the nanoreactor[38] or the choice of collective variables in metadynamics[78]). However, such enhanced sampling techniques restrict an exploration to one PES whereas our approach allows for an easy introduction of new reagents and reactants. Key strengths of our ansatz are (i) its agnosticism with respect to the molecular composition of the nanoscopic structure, which ensures general applicability, and (ii) overall algorithmic flexibility, which ensures seamless mix-and-match of structural models, interaction models, and optimization algorithms, due to its incorporation into the modular open-source SCINE software project for chemical reaction exploration[79,80] (see section 4.1 below for details). The overall concept and design of the QMG are summarized in Fig. 1.

We demonstrate how the QMG can be exploited as an enabler for two exploration strategies of atomistic nanoscopic systems. First, we show how automated reaction mechanism exploration algorithms can be leveraged in a routine fashion to search the reaction space in a biomolecule, including full consideration of potential side and degradation reactions. For this purpose, we present a workflow that automatically tames the combinatorial problem through truncation of the

# Quantum Magnifying Glass

**AUTOMATED MODEL GENERATION**

**EXPLORATION STRATEGY**

**Fig. 1 | Schematic workflow of interactive reactivity exploration in quantum mechanical/molecular mechanical (QM/MM) hybrid models.** The molecule shown is chain A of the peptide hormone insulin. The upper sequence of protocols (shown as blue boxes) illustrates the automated QM/MM model generation and subsequent interactive path generation and optimization protocol. The lower protocol sequence illustrates the automated generation of a QM model, followed by automated reaction search in this model, and subsequent QM/MM refinement. Relevant SCINE modules that drive the individual steps are depicted as purple gears. Employed electronic structure/ML/FF methods are depicted as white gears. All components of the modular SCINE software framework are accessible in the graphical user interface HERON, which includes, inter alia, a virtual environment[64] for immediate visual and (optionally) haptic[68–70] feedback denoted INTERACTIVE. READUCT is a module that provides direct access to structure optimization routines based on splines and curve optimization[85,86]. CHEMOTON[48] implements fully automated single-ended mechanism explorations of chemical reaction mechanisms based on first principles.

system size with a posteriori consideration of the nanoscopic environment. Second, we demonstrate how automated system-focused QM/MM hybrid model construction can overcome the limitation of model availability. The QMG then allows one to navigate intuitively through a multi-step catalytic reaction mechanism with interactive real-time calculations. The instantaneous feedback enhances chemical intuition while elucidating unknown reaction mechanisms and it significantly reduces the time required for the exploration.

## Results

We demonstrate the two possible algorithmic workflows depicted in Fig. 1 for two different chemical examples. Throughout this work, we rely on approximate semiempirical models for the structure generation in the reaction exploration, which allows for more extensive sampling, but can be replaced in our modular approach by more accurate electronic structure methods. We stress that the terms 'QM' and 'MM' in 'QM/MM hybrid model' can refer in principle to any two models of which the first one can describe bond formation and dissociation reactions and the second one can describe large nanoscopic structures. In order to distinguish the actually employed models from the concept of hybrid models, we refer to the structural model as a QM/MM model, whereas we refer to the specific potential-energy models as follows: the semiempirical model combined with SFAM is denoted SEQM/SFAM and the density functional theory model as PBE-D3(def2-SVP)/SFAM (see section 4).

### Automated QM/MM explorations through dimensionality reduction—esterification in insulin chain A

Since brute-force reaction exploration methods are computationally too demanding without being guided by some heuristic rules[43], it is desirable to evaluate chemical reactivity in a well-defined core model of a nanoscopic system and only subsequently assess the effect of the environment on the reactions found in the core model. In this way, much of the configuration space of a full nanoscopic system can be discarded during the exploration of (local) chemical reactivity. Important for this dimensionality reduction is that (i) the core model construction, (ii) the reaction mechanism exploration in the core

model, and (iii) the back-transplantation of the core model into the full atomistic structure can be fully automated. To implement this concept, we developed an exploratory workflow that

1. automatically determines a core model for the reaction (the **F**ocus step),
2. excavates a chemically valid subsystem from the nanoscopic environment (the **UN**tie step),
3. carries out an automated reaction search in this core model (the **N**avigate step),
4. transplants the reaction paths found back into the full system (the **E**xpand step), and
5. assesses the structural and energy effects exerted by the environment by refinement within the full QM/MM model (the **L**everage step).

We denote this sequence of **F**ocus **UN**tie **N**avigate **E**xpand **L**everage as FUNNEL, which represents an efficient approach for any nanoscopic system in which chemically relevant processes can be easily localized and isolated (as in active sites of enzymes). As an example, we analyze the esterification reaction of 1-propanol with the C-terminus of chain A of insulin. This reaction is an ideal candidate to assess the validity of FUNNEL for two reasons: First, the underlying esterification mechanism is chemically well-defined and can be encoded into a limited set of rules to be applied in the exploration protocol, hence keeping the search space comparatively small. Second, the relatively small structure allows us to also provide reference data for (i) a full QM reference exploration, and (ii) a single-ended QM/MM reference exploration (which both quickly become computationally unfeasible for larger structures). The esterification reaction is shown in Fig. 2A.

We first constructed our system-focused atomistic molecular mechanics model (SFAM)[31] and the corresponding QM/MM hybrid model[33] for chain A of the peptide hormone insulin. Our automated QM region selection algorithm applies a fragmentation scheme to construct multiple possible QM regions up to a maximally allowed size around some selected atom, which is supposed to be the local reaction center or close to it[33]. For each constructed QM region, the forces on

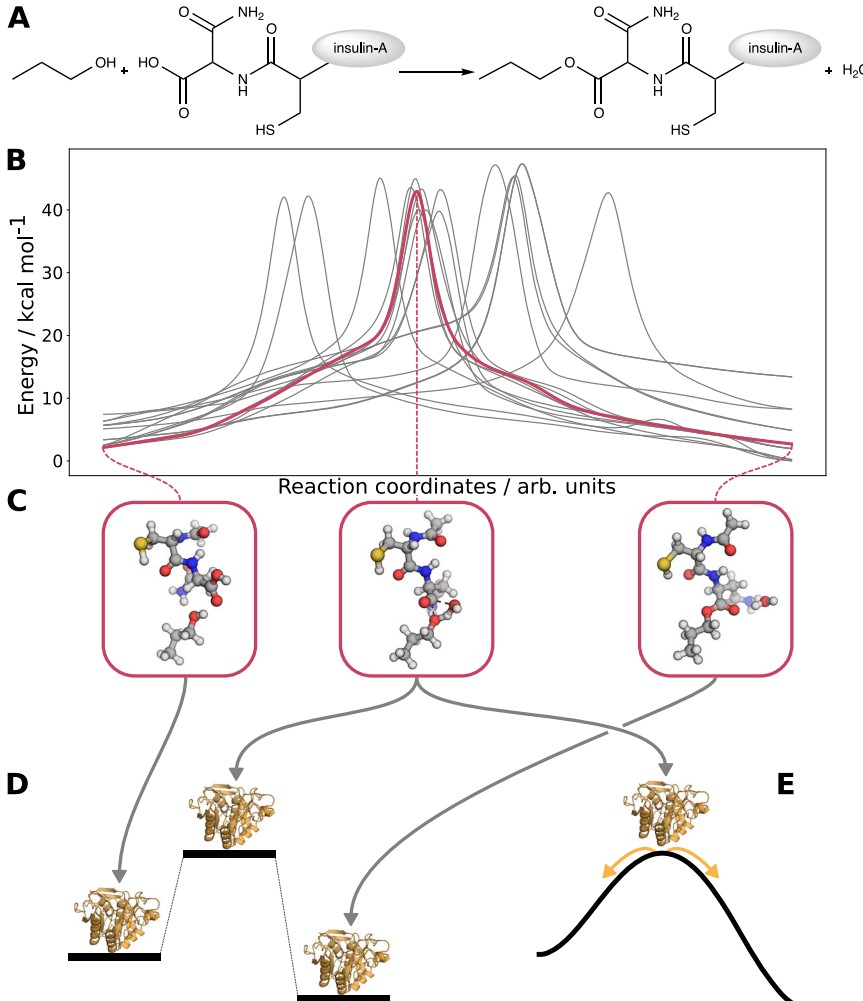

**Fig. 2 | The FUNNEL protocol. A** Reaction of 1-propanol with chain A of insulin (its backbone is represented by a gray circle). **B** B-splines fitted to the intrinsic reaction coordinates (IRC) of all esterification elementary steps discovered by autonomous exploration with the CHEMOTON program for the core model. Note that each of the elementary steps has a different reaction coordinate, so that no x-axis labeling can be given and the B-splines cannot be compared to one another along the x-axis. **C** Structures of the core model at the IRC end points and transition state for the elementary step with lowest reactant energy. **D** Transplantation of all stationary points of the core model into the full structural model environment and optimization of the environment with frozen core-model structures. **E** Alternative transplantation of only the transition state structure into the full structural model followed by a QM/MM IRC calculation. Source data of panel (**B**) are provided as a Source Data file.

atoms within a certain distance to the central atom are calculated and compared to the average of various larger reference models. The optimal QM region is then selected to be the one with the smallest average deviation from the reference calculations on the larger subsystems. For details on this algorithm see Ref. 33 and for the chosen settings see section 4. The optimal QM region based on our algorithm turned out to consist of 45 atoms.

We achieved a dimensionality reduction of the exploration workflow by focusing on a core model for the chemical reaction that corresponds to the QM region of the full QM/MM model. We then untied the core model from the backbone by dissecting $C(sp^3)$−C, $C(sp^3)$−N, and/or $C(sp^2)$−C bonds. That is, we restricted the set of dissectible bonds for the sake of simplicity, which may lead to larger QM regions and require organic ligands and substructures to facilitate the excavation of the core model (however, this may easily be generalized, if necessary).

We then navigated large sections of the reaction space of the excavated and valence saturated core model with CHEMOTON[48]. To accelerate this process, the initial exploration was carried out with semiempirical quantum chemical methods, for which we selected the

tight-binding model DFTB3[81]. In principle, recent developments of MLPs can replace expensive ab initio QM calculations, either by learning the difference between the results of an accurate calculation and a faster but less accurate method (denoted Δ-ML[82,83]) or by learning the ab initio PES directly[16]. We have contributed to both approaches with ultra-fast semiempirical calculations[67] and a lifelong MLP that can be trained continuously on all chemical elements[32]. For both approaches, the training can be focused either on a specific system or designed in such a way that the model can generalize across multiple different systems. Hence, they are not restricted to specific chemical elements and retain high accuracy. Therefore, both approaches are suitable for replacing approximate tight-binding methods in the future.

Since we focus on tracing the esterification reaction pathway, we limited the searched reaction space to bimolecular reactions (excluding dimerization reactions) and restricted the reactive sites to carboxylic acid and alcohol groups. This operator-induced steering was carried out by applying the STEERING WHEEL algorithm[49]. Note that the heuristic rules applied through the STEERING WHEEL were formulated as general as possible in order to minimize the truncation of

the search space. Therefore, we only defined (i) reactive sites, that is atom types in functional groups that are considered reactive, and determined the sampled reaction coordinates, and (ii) a total number of bond dissociation and formation reactions in the entire reactive complex. No atom-specific changes in the connectivity were enforced. By that, all reactions between carbon and/or oxygen atoms in a carboxylate group, and oxygen and/or hydrogen atoms in a hydroxyl group were tested. We emphasize that a major strength of our approach is the operator control of the depth of the exploration, which can be extended further to any degree desired, reducing uncertainties in the selected model and calculated properties. The exploratory bias arising from the choice of reaction rules is a source of aleatoric uncertainty. By continuously expanding the set of applied heuristic rules and therefore increasing the exploration breadth, the bias can be decreased systematically. Even in the case of very extensive sampling, epistemic uncertainties can never be fully eliminated (see our discussion in Ref. [14]). Therefore, our framework allows for a flexible exchange of the underlying QM and MM model, and therefore refinement of a fast, but approximate exploration with more accurate models. The increased requirement of resources can be reduced by applying more accurate models only on a few elementary steps that are selected by a Gaussian Process model[84].

The aleatoric uncertainty, by contrast, may be reduced by sampling more trajectories. This can be achieved directly in the FUNNEL algorithm due to the initial elementary step search with CHEMOTON which provides multiple parameters (such as the number of sampled conformers of each reactant, number of rotamers in bimolecular reactive complexes, and the number of attack points) that can be adjusted in order to explore more trajectories. With an increased number of trajectories, more conformational degrees of freedom of the stationary points are scanned. Therefore, the likelihood increases that the path associated with the lowest overall reaction energy and/or barrier was covered, and hence, the uncertainties of both reaction and activation energies are reduced. The number of reactive sites, the allowed number of bond formation reactions, dissociation reactions, and attack points unequivocally define the reaction space to be searched in the exploration, and these are kept fixed throughout this study.

The defined reaction rule set prepared 510 separate reaction trials, consisting of approximately $10^6$ individual energy calculations which required 24 h of computing time. Hence, the whole procedure was amenable to a standard desktop computer. Our exploration identified 17 elementary steps of the single-step esterification given in Fig. 2A/B, among 103 elementary steps found in total. Elementary steps connect (sets of) structures either via a single transition state (first-order saddle point on the PES) or a barrierless association or dissociation reaction. A structure is a stationary point on the PES that corresponds to a three-dimensional arrangement of nuclei with a given molecular charge and spin multiplicity. Structures that are local minima on the PES can be grouped into compounds if they share the same nuclear composition and connectivity, but differ in the three-dimensional arrangement. Accordingly, elementary steps can be grouped into reactions, if the associated reactant and product structures belong to the same reactant and product compounds. For details on these definitions, we refer to Ref. [42]. The 103 elementary steps were grouped into 18 reactions (for a detailed description, see the Supporting Information). One of the 18 reactions corresponds to the initial step of a two-step esterification mechanism, yielding a tetrahedral intermediate. Given that our heuristic reaction rules were defined in a general manner in terms of functional groups, we were able to extend the exploration straightaway with unaltered reaction rules allowing for unimolecular reactions of the kinetically favored products of the exploration up to that point. This extension of the exploration completed the two-step mechanism. Given that its activation energies were similar to the one-step mechanism, we focus on the discussion of the latter. For details on the two-step mechanism, see section 4.2 of the Supporting Information.

Throughout this manuscript, we will refer to identical elementary steps if the same transition state was selected or the same reactive complex and reaction coordinate were sampled. To justify the initial structural confinement and choice of the QM core structure, we verified whether the elementary steps identified within the core model structurally align with those that would emerge from an exploration in a QM model of the full structure. Automated explorations employing a QM model on the full structure are generally not feasible in terms of the computational effort involved. Moreover, optimizations of transition states on a shallow PES of a system with more than 300 atoms are often prone to failure. For our insulin example, only two of the 17 elementary steps could be successfully optimized in the full model. These limitations illustrate why it is both reasonable and necessary to carry out vast explorations preferably on truncated structural models. For the two successful elementary steps, we found a huge spread in energies (the difference in reactant energies is 33.4 kcal mol⁻¹ compared to 1.2 kcal mol⁻¹ in the QM core model), indicating that hybrid models are required to accurately model interactions between the reaction center and the environment, while ensuring sufficient sampling of the large conformational space. For the elementary step with the lowest-lying reactant energy in the full QM model, the root mean square deviations (RMSDs) between the QM core structure and the identical atoms in the full QM structure (excluding hydrogen atom positions) amount to 1.0 Å (reactant), 0.8 Å (transition state) and 1.4 Å (product), and can primarily be associated to rotation and rearrangement of the carbon backbone of propanol and the water molecule in the structurally unconstrained core model in reactant and product. However, the rest of the structure, especially the reactive atoms and their immediate structural surroundings, maintains structural integrity compared to the full model, which is a good indicator that the QM core model structure is well chosen.

After identifying these 17 reaction paths in the core model, we then assessed the structural and energy effects of the protein environment by transplanting the core-model stationary points back into the full QM/MM structural model. The reinsertion is implemented by a quaternion fit of the anchor point atoms of the core model to their corresponding positions within the full structure model. The anchor points are defined as the atoms in the QM region that are covalently bound to an atom in the MM region. Because the quaternion fit represents a three-dimensional rotation, at least four non-collinear points in space (i.e., the anchor points) are required to uniquely determine these rotations. If there are fewer than four anchor points, the algorithm will recursively extend the anchor points by adding their bonding partners (except valence-saturating hydrogen atoms, which have no analog in the full QM/MM model) until at least four atoms are selected.

For the expansion step, we developed two strategies. In the first strategy, reinsertion was carried out for all stationary points of the core model network; that is, reactant, product, and transition state structures were refined in the QM/MM model to yield SEQM/SFAM stationary points (Fig. 2D). In the second strategy, only the transition state structure was reinserted, from which the intrinsic reaction coordinate (IRC) can be calculated, delivering a complete SEQM/SFAM minimum energy path (MEP) (Fig. 2E).

These two strategies will offer complementary advantages if one wants to leverage how the environment affects the isolated reaction. In D of Fig. 2, structure optimization of only the environment region conserves the core stationary points and enables one to assess environment effects on the relative energies of stable (core-model) intermediates and transition state structures. In E of Fig. 2, by contrast, an optimized SEQM/SFAM MEP is calculated by optimizing a SEQM/SFAM transition state and subsequently relaxing both QM and MM degrees of freedom in an IRC scan. While strategy E yields the SEQM/SFAM MEP

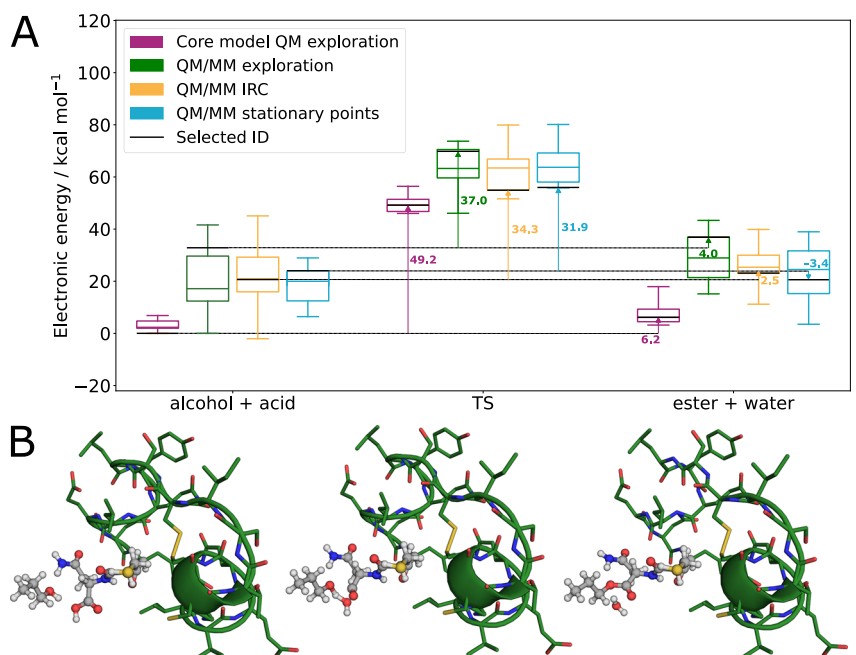

**Fig. 3 | Reaction paths of the esterification in insulin. A** Box plots for the spread of relative electronic energies of the stationary points of the reaction for the core model (*n* = 17), the 'quantum mechanical/molecular mechanical (QM/MM) stationary points' of the transplanted core model into the full structure with environment relaxation (*n* = 17), 'QM/MM IRC' obtained from an intrinsic reaction coordinate (IRC) scan starting from the transition state in the full QM/MM model (*n* = 16), and 'QM/MM exploration' calculated in a single-ended reaction search in the full structural model starting from the reactive complex of the reactants (*n* = 9). The energies were calculated with PBE-D3(def2-SVP)/SFAM single-point calculations on DFTB3/SFAM structures (SFAM: system-focused atomistic models), and are given relative to the lowest reactant conformer energy of the core model and of the single-ended QM/MM strategy for the three QM/MM strategies. The whiskers of the boxes show the minimum and maximum values, the boxes depict the second and third quartile, and the line in the box denotes the median of the energy values. The horizontal bar annotates the elementary step with the lowest reactant energy in the core model, and the respective elementary step in the QM/MM model for which the core model step served as an input. The activation and reaction energies for this elementary step in all approaches are annotated with vertical arrows (for details and energy decomposition analyses, see Table 8 in the SI). **B** Stationary points of one of the DFTB3/SFAM minimum energy reaction paths; the QM region is shown in ball-and-stick representation, whereas the MM region is presented in a stick-cartoon representation. Hydrogen atoms are omitted in the MM region for the sake of clarity. Source data of panel (**A**) are provided as a Source Data file.

directly from the core model exploration with a single Hessian calculation, strategy **D** yields SEQM/SFAM stationary points from the relaxation of the environment alone. Generally, strategy **D** is computationally less demanding than **E**. However, the absolute computational costs of these strategies depend on the selected embedding scheme and Hessian matrix calculations. Throughout this work, we applied the partial Hessian scheme focused on the QM region only (optimizing the MM region to an energy minimum for each change in the QM region) as described in the Supporting Information. This makes full SEQM/SFAM transition state optimization feasible. Note that the effective Hessian approach allows for more accurate calculations of harmonic frequencies through the inclusion of QM-MM interactions into the QM Hessian calculations in the form of quadratic couplings[60]. However, this requires diagonalization of full Hessian matrices in every macroiteration step of a (transition state) optimization, including QM/QM, QM/MM, and MM/MM interaction terms, which we omit in this study for computational feasibility reasons. For insulin, we chose a mechanical embedding scheme due to its low computational costs and straightforward interpretability when comparing the expansion strategies **D** and **E**. For more information on the applied embedding schemes, we refer to section 4.

To determine the reliability of the expansion step in FUNNEL for approaches **D** and **E**, data from a full single-ended QM/MM exploration serve as a reference. This approach, although conceptually intuitive for the study of reactions in nanoscopic systems, is computationally much more expensive due to its single-ended nature, involving potentially thousands of single-point and Hessian calculations. Approach **E**, by contrast, can simply be started from a transition state structure (which

is often difficult to find), and is therefore computationally cheap to apply.

Due to the conformational variety of transition states in elementary steps that belong to the same chemical reaction, a range of activation and reaction energies is obtained. This is a major strength of our automated approach, as the increased sampling of reactive trajectories increases the probability that the key low-energy paths are identified. We show a range of relative electronic energies in Fig. 3A for the core model and the two expansion strategies calculated with PBE-D3(def2-SVP)/SFAM single-point calculations on DFTB3/SFAM stationary points. We find excellent quantitative agreement of medians (line in the box of the box plot) of the energy value for reactant, transition state, and product between approaches **D**, **E**, and the QM/MM exploration. This demonstrates that both QM/MM IRC and QM/MM stationary points deliver qualitative reaction trajectories on the QM/MM PES, at much reduced computational costs compared to the full single-ended exploration. Overall, the effect on the energies within the nanoscopic system compared to the core model is considerable as the spread is much more pronounced (minimum–maximum range of whiskers of the box plots). This highlights that a QM/MM description of the full system can be essential for reliable reaction mechanism elucidations.

To demonstrate the effect of the QM/MM embedding on local reactivity for both strategies **D** and **E**, we discuss energy trends of the minimum energy path with the lowest reactant conformer energy in the core model highlighted in Fig. 2A; the corresponding structures of the QM/MM IRC are shown in Fig. 3B (for detailed results, we refer to the Supporting Information). The reaction energy $\Delta_R E$ of the isolated

core model is + 6.2 kcal mol$^{-1}$, and the corresponding activation energy $E_A$ is 49.2 kcal mol$^{-1}$. Among the QM/MM approaches, we find a consistent lowering of the activation energy, being 31.9 kcal mol$^{-1}$ for approach **D**, 34.3 kcal mol$^{-1}$ for approach **E**, and 37.0 for the QM/MM exploration reference.

Although **D** does not provide a SEQM/SFAM MEP for this reaction, it can be helpful to specifically analyze the magnitude of individual interactions between a local minimum of the QM region and its environment. For instance, the decrease of $E_A$ of 17.3 kcal mol$^{-1}$ in the activation energy can be attributed mainly to strong attractive dispersion interactions between the QM and MM subsystems ($-18.4$ kcal mol$^{-1}$). Compared to that, the Coulomb interactions and Pauli repulsion in our SFAM force field (see Eq. (7) and (10) in Ref. [31]) are relatively weak ($-4.4$ kcal mol$^{-1}$ and + 5.6 kcal mol$^{-1}$). We find that the structural changes in the MM region are overall more pronounced for the QM/MM stationary points approach compared to the QM/MM IRC, which is a reasonable result as the quantum region is kept fixed. Therefore, the QM/MM IRC approach provides a more reliable representation of a QM/MM reaction path, which is reflected in better quantitative agreement in calculated activation and reaction energies compared to the QM/MM reference exploration.

Hence, both strategies, **D** and **E**, for the calculation of QM/MM reaction energies and barrier heights allow us to assess the generic energies of the core model (considered by the structure of the isolated core embedded in the protein environment in **D**) compared to the adiabatic energies from the QM/MM IRC data.

## Real-time interactive QM/MM – Hydrogenation of propylene catalyzed by a HKUST-1 analog

If a core model is difficult to extract (e.g., when multiple transition metal atoms are to be taken into account) and/or involved chemical transformations are to be expected (e.g., in a multistep catalytic cycle), automated procedures become inefficient without human intervention. Furthermore, hybrid models are scarce for non-biomolecular compounds. For such situations, we introduce an approach that combines our system-focused hybrid model[33] with real-time quantum chemical calculations[63]. In this example, the exploration bias did not result from the automated application of predefined reactivity rules as in the FUNNEL approach, but from the interactive nature of the exploration, where only the reaction paths interactively probed by the operator were evaluated.

Real-time calculations produce electronic energies and gradients (i.e., forces) at such high frequency that they allow for instantaneous processing by an operator. Since the inspection of alpha-numerical output would require too much time for a human to process in such a setting as they are produced at very high pace, it is convenient to have visual (on screen energy monitoring) and haptic (force feedback) presentations. While visual feedback requires an update rate of approximately 25 Hz for smooth real-time experience, haptic feedback (a physical force feedback upon molecule manipulation) demands a higher update rate due to the higher sensitivity of the human tactile sense. Throughout this work, we focus on real-time explorations with a mouse-based input, allowing for the aforementioned update rate of about 25 Hz in order to achieve a real-time experience. We found that mouse-based interactions with HERON provided robust and efficient means for intuitive interactive explorations.

The real-time feedback offers various advantages for navigating the complex reaction space of nanoscopic systems. For example, the manipulation of three-dimensional structures with simultaneously carried out calculations allows one to generate approximate minimum energy paths[85,86], because those parts of the structure that are not manipulated are continuously relaxed by rolling structure optimization. As the human sense is trained on at most three-dimensional ($\mathbb{R}^3$) objects, it is generally counterintuitive and challenging to the human visual sense to perceive high-dimensional spaces. For instance, the 3$N$ degrees of freedom of an $N$-atom molecule span a $\mathbb{R}^{3N}$ space of possible reaction coordinates. As an $N$-atom molecule is represented in an $\mathbb{R}^3$ space in HERON, it becomes intuitive for the operator to grasp and manipulate structures interactively, and generate target-specific reaction paths within the $\mathbb{R}^{3N}$ space. These paths are then handed over to automated algorithms for the optimization of QM/MM MEPs (the graphical user interface in which we implemented this setting is depicted in Fig. 4). As a result, the complete characterization of a catalytic cycle combining interactivity and automated MEP search algorithms can be accomplished within several minutes on a standard workstation.

We first demonstrate how a QM/MM hybrid model can be interactively constructed for a non-biomolecular system. As an example, we selected a rhodium-copper analog of HKUST-1, which is a $Cu_3(btc)_2$ (btc$^{3-}$ = benzenetricarboxylate) MOF[87]. In the paddle-wheel-like local structures in this nanoscopic system (shown in Fig. 5), copper acetate moieties coordinate to benzenetricarboxylate linkers. One coordination site at the metal center is vacant, which is where an incoming ligand can bind. Its catalytic reaction of a hydrogenation reaction has been studied experimentally and computationally[88,89]. The in-situ Raman spectroscopy and diffuse reflectance infrared Fourier transform spectroscopy experiments[89] show that a propyl ligand is formed during the reaction, that hydroxyl groups exist in the structure (albeit already before the hydrogenation reaction due to defects arising from the synthesis of the MOF), and that rhodium-oxygen bonds are broken during the hydrogenation reaction.

Based on these experimental insights and complementary density functional theory (DFT) calculations of a truncated small-structure model consisting of the two metal centers and four acetyl groups, hence neglecting the remainder of the MOF environment, Chen et al.[88,89] postulated the catalytic cycle to be initiated by $\eta^2$-coordination of propylene at the vacant site of Rh$^{2+}$. This is then followed by the association of $H_2$ and cleavage of the $H_2$ bond and a metal-oxygen bond to yield a metal hydride, Rh−H, and a hydroxyl, HO−R, species. This is followed by transfer of the metal−H hydride to propylene and product formation by transfer of the OH-hydrogen atom (shown in Fig. 5, right). Because the activation energies of the calculated reaction path did not agree with experimental observations, Chen et al. also proposed a second catalytic cycle[89] in which the second hydrogen atom required for hydrogenation is transferred from a second (adsorbed) $H_2$ molecule, bypassing the hydrogen transfer from a carboxylate group.

Because the structural model considered in the work of Chen et al.[88,89] lacked electrostatic interactions between the reactive site and the MOF scaffold as well as the overall rigidity of the large framework, which might be crucial for quantitatively understanding catalytic properties of the MOF, we decided to study this reaction with our interactive SEQM/SFAM approach.

Our structural model was constructed to be an extended fragment of the bimetallic node that is composed of the secondary building unit (*cf.*, Fig. 5) employed for parametrization of the SFAM model (see Section 4) and four additional benzenetricarboxylate moieties. Starting from this structure, we constructed a full-fledged QM/MM structural model of all intermediates along the catalytic cycle in a straightforward and efficient way interactively within HERON with GFN2-xTB/SFAM as the electronic structure model. These intermediates were then subjected to double-ended reaction paths searches in order to characterize transition states and generate MEPs connecting them (for details on this procedure, see Ref. [85]). The reaction and activation energies were then refined with M06-L(def2-TZVPP)/SFAM single-point energy calculations.

One huge benefit of our ansatz is the efficiency with which SEQM/SFAM MEPs can be found. The automated transition state optimizations and IRC calculations of our interactively generated guess structures were carried out in a few hours of computing time on a standard

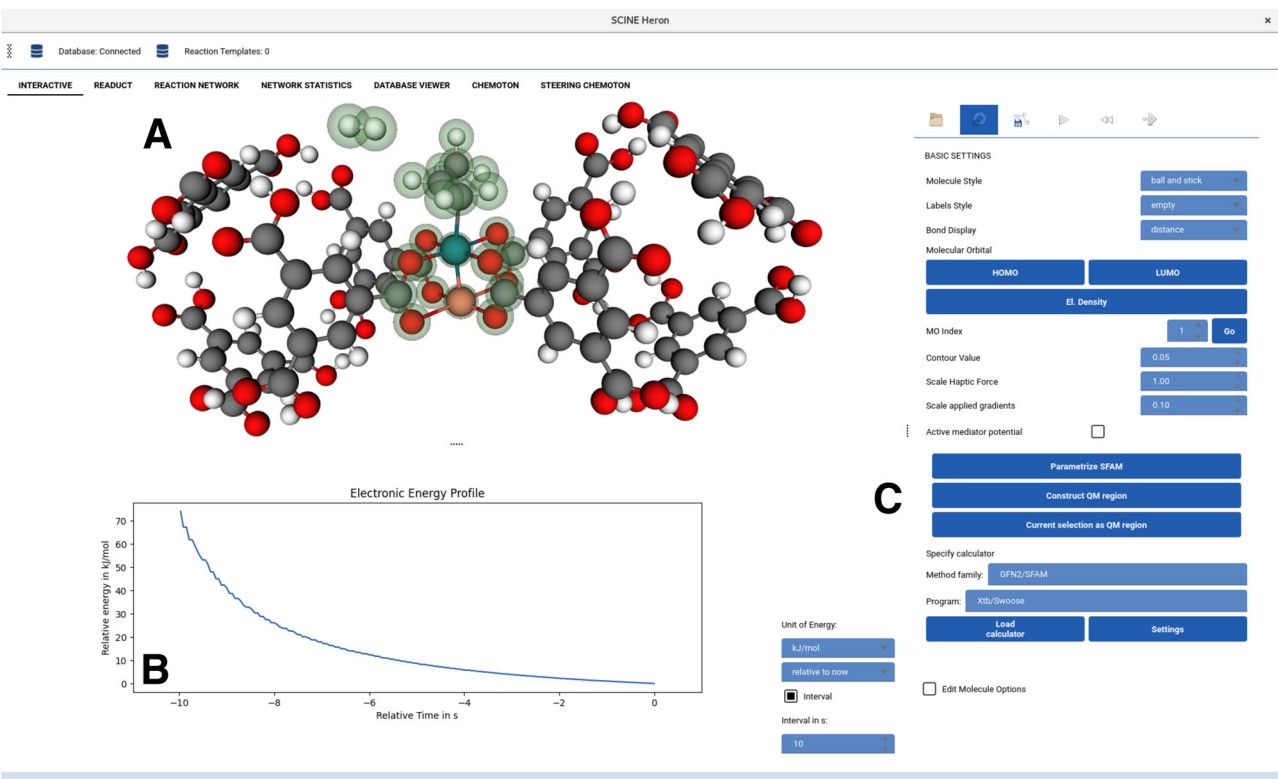

**Fig. 4 | Screenshot of the graphical user interface HERON, which implements the QMG interactive exploration of reaction paths for subsequent (non-interactive) automated refinement. A** Molecular complex of propylene, molecular hydrogen, and the active site of the MOF catalyst HKUST-1 with one copper nucleus replaced by rhodium. The green transparent spheres mark the selected QM region in the quantum mechanical/molecular mechanical (QM/MM) model. Real-time structure manipulations can be carried out directly within this window steered either by a computer mouse or by a haptic device. **B** The electronic energies associated with every new structure produced are continuously monitored. **C** The system-focused atomistic model (SFAM) force field can be parametrized within HERON and the QM region can be changed on-the-fly, either by manual selection or by the automated algorithm introduced in Ref. 33 and applied in section 2.1.

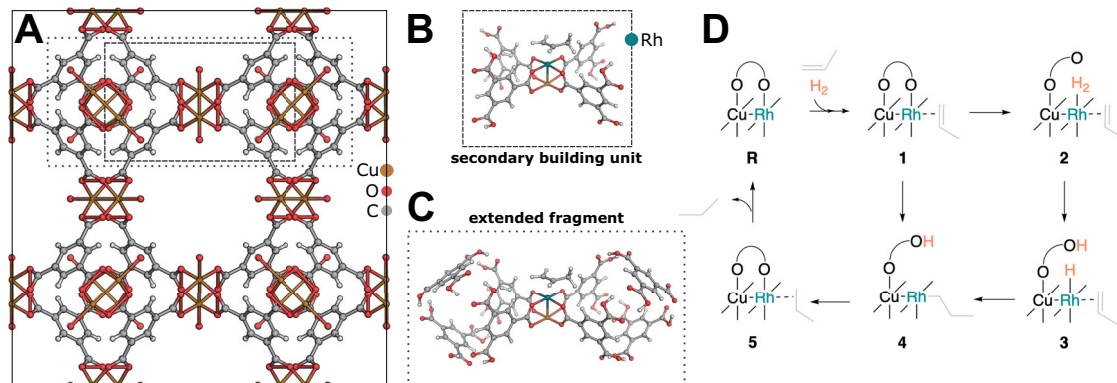

**Fig. 5 | Structural models for HKUST-1 and key reaction steps. A** HKUST-1 unit cell; (**B**) tetracarboxylate secondary building unit of a HKUST-1 analog featuring a rhodium-(II)-copper-(II) bimetallic node; (**C**) the corresponding extended fragment employed for exploration; and (**D**) the proposed catalytic cycle[88] at the rhodium-copper active site for the hydrogenation of propylene.

desktop computer for the different SEQM/SFAM reaction paths shown in Fig. 6. Activation and reaction energies were then refined with DFT/SFAM single-point energy calculations. The efficiency of this exploratory strategy allowed us to systematically probe different reactive sites within the system and draw an extensive picture of the complex reaction network under full consideration of different possible reaction paths between coordination isomers. For instance, we compared both axial and equatorial $\eta^2$-coordination of $H_2$ to the rhodium ion, probed for hydrogen transfer to any of the four Rh-coordinating

oxygen atoms (steps **2 → 3** in Fig. 6), and explored the hydrogenation reaction via an 1-propyl and 2-propyl intermediate.

We started the interactive exploration from compound **1**. Propylene was coordinated in an $\eta^2$-manner to rhodium, and the propylene double-bond was aligned in parallel to one O–Rh–O axis (highlighted in orange in Fig. 6). In a first step, the $\eta^2$-coordination of $H_2$ was kinetically favored for the equatorial isomer (**2$^{eq}$**) compared to the axial isomer (**2$^{ax}$**). In both cases, the $H_2$ coordination was concerted with carboxylate decoordination through scission of one Rh–O bond. $H_2$ is

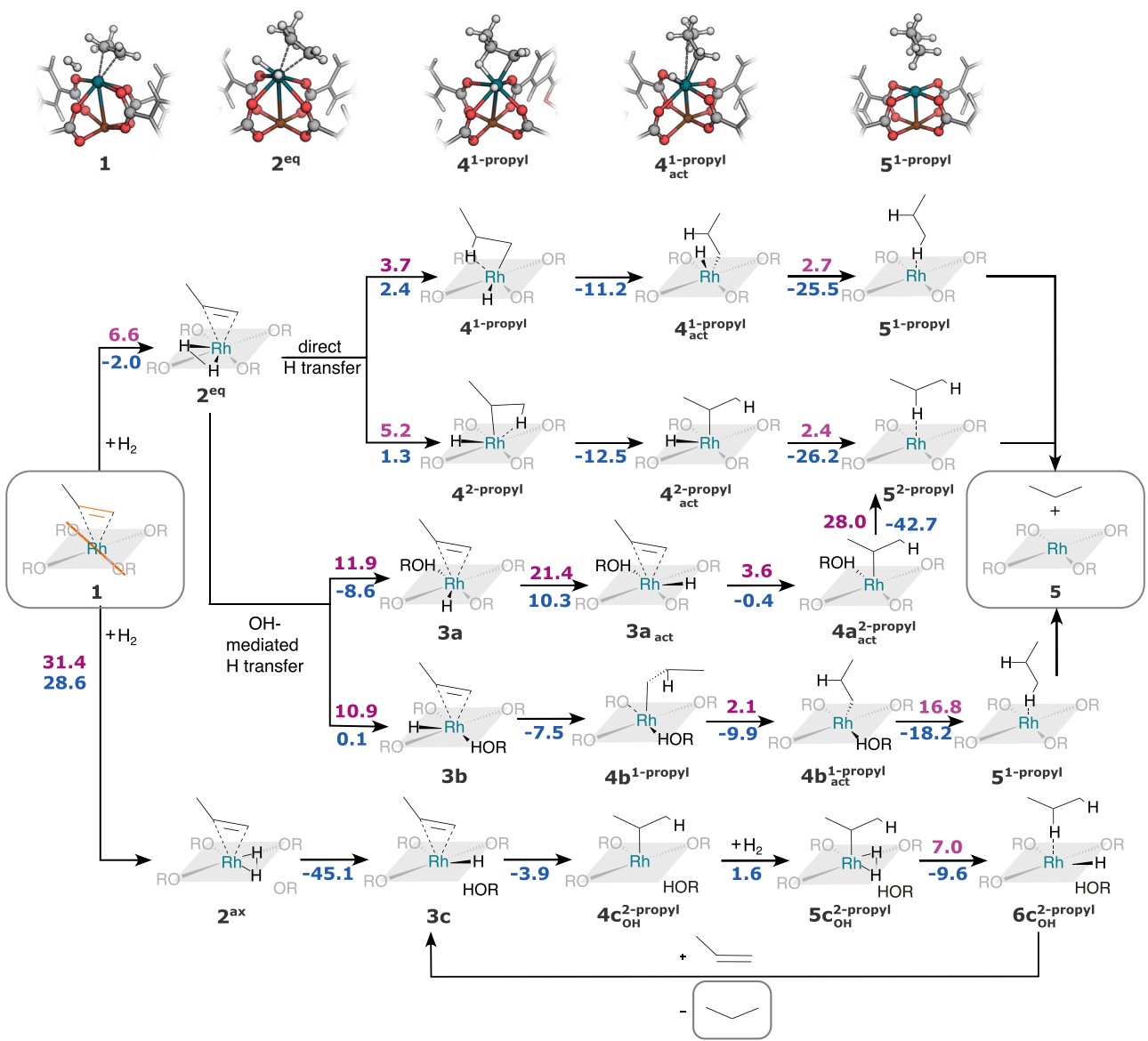

**Fig. 6 | Relative electronic energies of the intermediates found by real-time exploration and subsequent (non-interactive) double-ended transition state searches with GFN2-xTB/SFAM as the hybrid model.** The relative M06-L(def2-TZVPP)/SFAM (SFAM: system-focused atomistic model) energies are given in purple for activation energies and in blue for reaction energies. Cut-outs of the molecular structures are shown for the upper-most reaction path. The explored structure was the extended fragment depicted earlier in Figs. 4 and 5. The orange lines in compound **1** highlight the collinearly aligned bonds in the catalyst and bound substrate. Reactant and product compounds are in gray boxes.

located above the Rh–O plane that was highlighted in gray in Fig. 6, and propylene was adsorbed at 2.1 and 2.2 Å distance. From species **2^eq**, we found reaction paths for direct hydrogen atom transfer involving consecutive cleavage of the Rh–H bonds (top two paths in Fig. 6) without formation of an OH species. This sequence of elementary steps has not been discussed in previous work[88,89]. Depending on which hydrogen atom was transferred first to the substrate, a 1-propyl or 2-propyl intermediate was formed as a reaction intermediate (**4^1-propyl** or **4^2-propyl**, respectively). In both cases, the hydrogen atom could only be transferred to the substrate, if it was aligned collinear to the propylene double bond. The propyl ligands in the intermediates **4^1-propyl** and **4^2-propyl** then reoriented in a barrierless fashion to yield **4_act^1−propyl** and **4_act^2−propyl**, respectively. In these species, the propyl ligand was bound in an $\eta^1$ fashion to the rhodium center. The second hydrogen atom was located perpendicular to the C-C axis of the former double bond, and it was transferred to the substrate with an activation energy of 2.7 and 2.4 kcal mol⁻¹ for the 1-propyl and 2-propyl

intermediates, respectively. This step also involved re-coordination of the oxygen atom and full re-establishment of the metal site environment. Overall, both pathways yield the same product. The molecular structures of intermediates of the 1-propyl reaction path are shown in the top panel of Fig. 6.

By contrast, reaction paths that proceed from **2^eq** to OH-intermediates are kinetically disfavored, compared to the direct hydrogen atom transfer in our QM/MM model even under consideration of the full isomer space. We located transition states for the protonation of two of the four carboxylate groups (compounds **3a** and **3b**), the other two groups were excluded due to steric constraints imposed on the second hydrogen atom transfer in subsequent steps. We studied reaction paths starting from the two isomers **3a** and **3b**, with different carboxylate groups protonated. Species **3a**, which is the starting structure for 2-propyl-type hydrogenation on the OH-mediated branch of the given network, first undergoes H-activation to yield **3a_act** to align the Rh–H bond with the double-bond axis. This

step exhibited a barrier of 21.4 kcal mol$^{-1}$. For species **3b**, however, the Rh−H was in an activated configuration, and could be transferred to the substrate in a barrierless step to yield a 1-propyl-type intermediate **4b$^{1\text{-propyl}}$**. After substrate reorientation to **4b$_{act}^{1-propyl}$**, the second hydrogen atom transfer was highly exothermic (− 18.2 kcal mol$^{-1}$), but with a barrier of 16.8 kcal mol$^{-1}$.

Therefore, we conclude that the reaction path via OH-formation is disfavored from an equatorially coordinated H$_2$ intermediate in our model. Additionally, we studied OH-mediated hydrogenation from intermediate **2$^{ax}$**, i.e., for a different Rh−H$_2$ isomer that was studied previously[88,89]. Initial hydrogen coordination was endothermic with a barrier of 31.4 kcal mol$^{-1}$ in our model. The mechanism involving **4$^{1\text{-propyl}}$** or **4$^{2\text{-propyl}}$** could not be found starting from compound **2$^{ax}$**. Instead the formation of the OH-species **3c** and subsequent proton transfer to yield **4c$_{OH}^{2-propyl}$** were exothermic (− 45.1 and − 3.9 kcal mol$^{-1}$) and barrierless. For species **4c$_{OH}^{2-propyl}$** we found a kinetically favored mechanism involving coordination of a second H$_2$ to the vacant site at the rhodium ion compared to direct hydrogen transfer from the decoordinated, protonated carboxylate group, which is in agreement with previous work[89].

In summary, we found the proposed path[89] as well as a different reaction path favored in our model. For both mechanisms, the initial H$_2$ association reaction showed the largest activation energy (6.6 kcal mol$^{-1}$ in our path), which is in agreement with previous studies[88,89]. We achieved quantitative agreement with the experimentally determined activation energy of 6.3 kcal mol$^{-1}$. Moreover, our model of a second catalytic cycle involving the coordination of a second H$_2$ molecule featured the correct activation energy of 7 kcal mol$^{-1}$. In addition, both mechanisms involve a direct hydrogen atom transfer from the rhodium ion with the protonated carboxylate group in a spectator role stabilizing the decoordination from (and hence, activation of) the active site. Given that the experimental results[89] point towards breaking of the rhodium−oxygen bond, a propyl intermediate is formed, and hydroxyl groups exist already before and during the hydrogenation reaction, we conclude that our mechanisms are in accord with experiment and that the hydrogenation reaction most likely takes places via direct hydrogen atom transfer from the rhodium ion with protonated carboxylate groups stabilizing vacant active sites.

## Discussion

The quantum magnifying glass (QMG) enables an in-depth understanding of complex reaction paths within nanoscopic systems at the atomistic level. We analyzed an esterification reaction in insulin and the multi-step hydrogenation of propylene catalyzed by a metal-organic framework (MOF), highlighting how the study of diverse composite compounds can benefit from system-focused molecular mechanics models, which can be automatically constructed and validated.

The esterification reaction in insulin, for which no experimental data are available to compare, served the purpose of showing how efficient reaction exploration can be accomplished by exploring first the generic reactivity of a small model and by subsequent transplantation into the full structural model, which affected calculated reaction energies significantly as is known well from QM/MM studies[90]. Furthermore, we showed that interactive multiscale models are capable of efficiently finding the elementary steps postulated for the propylene hydrogenation by a MOF catalyst, along alternative pathways not yet discussed in the literature. The transferability of the QMG to more complex enzymatic reactions needs to be carefully assessed. Especially the exploration of multi-step catalytic cycles involving transition metal ions can present significant challenges to the electronic structure model employed. Since ultra-fast SEQM/MM models trade speed for accuracy, more accurate electronic structure methods will be required for refinement, but at the expense of significantly increased computational resource requirements. Although this can be done in an automated fashion in the background then, it requires an effort that is beyond the scope of the present work.

The QMG is not limited to the examples shown here, since our implementation unifies all steps of model construction and simulation in a single framework that offers a high degree of flexibility and allows a mix-and-match of structural models, electronic structure theories, and exploration strategies. The combination of automated algorithms with real-time feedback offers opportunities to study the reactivity of complex chemical systems with little expertise and technical knowledge.

Hence, the QMG, as a concept for exploring, understanding, and even designing reaction mechanisms in nanoscopic systems, harbors the potential of becoming a canonical instrument for reactivity studies at the nanoscale, facilitated by the free availability and low entrance barrier of our implementation.

## Methods

### Conceptual and software considerations

The interactive and automated exploration designs of this paper have been implemented in the SCINE[79] software framework developed within our group (see the Supporting Information for references to all SCINE modules (these are, READUCT, SWOOSE, CHEMOTON, PUFFIN, HERON, MOLASSEMBLER, SPARROW, and INTERACTIVE) which we distribute free of charge and open-source). The accessibility of our algorithms is achieved through an interface of all relevant SCINE software modules with the SCINE graphical user interface HERON. SCINE INTERACTIVE allows one to intuitively generate structures and trajectories that are a suitable input for further stationary point or MEP search algorithms. We implemented a new graphical interface to READUCT[85], which provides direct access to all structure optimization routines implemented in READUCT. For QM/MM (transition state) structure optimizations, the protocol described in the Supporting Information can be applied. In this way, HERON allows general structure modifications with any backend program supported by the SCINE framework and therefore enables a seamless structure- and model-transfer between INTERACTIVE, READUCT, and CHEMOTON.

HERON allows for automated generation of the SFAM molecular mechanics model in real time as long as a semiempirical reference method is employed for ultra-fast reference data generation. Available ultra-fast methods comprise the NDDO, the DFTB and the xTB semiempirical method families. Standard quantum chemistry packages can also be employed. Although the higher computational cost of standard electronic structure methods such as DFT hamper interactivity of the exploratory process, they can be leveraged in subsequent refinement.

The interactive generator of the QM/MM hybrid model in HERON offers three different options for the selection of a QM region: (i) a physically motivated QM region selection based on the formalism presented in Ref. 33 and discussed in the Supporting Information, (ii) a structure-based QM region selection that delivers a spherical fragment around a central atom of choice, and (iii) a manual selection of a set of nuclei that then constitute the QM region. Because the parameters of the SFAM molecular mechanics model are generated automatically for the complete system, a redefinition of the QM region is possible at any time during a QM/MM exploration. The physically motivated selection algorithm (i) can also be carried out in real time with ultra-fast semiempirical reference methods[65,67].

Note that the overall design concept and the modular character of the SCINE software framework enables seamless "mix-and-match" of different types of exploration strategies (brute-force, guided by encoded rules and reaction filters within CHEMOTON, and optimization of a single reaction path with READUCT) with different structural models (core system, nanoscopic system) and electronic structure models (QM, semiempirical QM (SEQM), MLP, MM, QM/MM) as conceptualized in Fig. 1.

## Computational methodology

All data management, quantum chemical calculations, and structure manipulations were conducted within SCINE[79]. The automated reaction exploration of the esterification reaction was carried out with CHEMOTON and the Steering Wheel algorithm[49] which allows us to steer the exploration toward our target reaction within HERON. All reaction trials were carried out with the default settings of CHEMOTON 3.0 and the Newton Trajectory 2 algorithm[48]. All calculations were carried out with a PUFFIN singularity container on high-performance computing infrastructure or with a local installation of the electronic structure programs whose versions are given below. The molecular graphs (required for sorting all found chemical structures into compounds and flasks) were constructed with the library MOLASSEMBLER[91].

All electronic structure calculations were carried out by the SCINE Calculator interface that allows one to separate any structure manipulation from the electronic structure model for obtaining energies and gradients for a chemical structure. The PM6 and DFTB3 calculations were carried out with SPARROW[67], the GFN2-xTB[92] calculations were carried out with xtb 6.5.1[93] interfaced to our software framework. For all DFTB3 calculations, the 3ob-3-1 parameter set[94] was employed. All molecular mechanics calculations were based on the SFAM model[31] and executed with SWOOSE. The DFTB3/SFAM calculations applied mechanical embedding and the GFN2-xTB/SFAM calculations applied electrostatic embedding based on SFAM point charges at the positions of the classically described nuclei. The charges close to the QM/MM boundary were redistributed based on a scheme that is described in Fig. 2 of the review article by Senn and Thiel[95]. The DFT reference calculations for the SFAM reference data generation were based on the Perdew-Burke-Ernzerhof (PBE)[96] exchange-correlation density functional with D3 dispersion correction[97] and Becke-Johnson damping[98]. Single-point calculations of the stationary points of the SEQM/SFAM reaction path of the esterification of insulin and of the propylene hydrogenation in HKUST-1 were carried out with the PBE and the Minnesota M06-L[99] exchange-correlation density functional, respectively. All DFT calculations were carried out with TURBOMOLE 7.4.1[100] with the def2-SVP basis set for insulin and the def2-TZVPP basis set for the HKUST-1 analog[101]. Density-fitting resolution of the identity was exploited with the def2/J auxiliary basis set[102]. The ground state spin of the HKUST-1 analog was found to be a triplet configuration with M06-L. Note that a low-spin (singlet) solution had to be adopted for the approximate GFN2-xTB tight-binding model, which is best practice for this model as it does not contain a spin-dependent energy contribution[93].

## QM/MM model for insulin

The starting structure for chain A of (dry) insulin was extracted from a crystal structure (PDB-ID: 1AIO[103]) and automatically processed with the ASAP protocol[34] for structure preparation. For the protonation pattern in insulin determined by ASAP, we assumed a pH value of 7, which enforced charge neutrality of the thiol group in the vicinity of the reaction center. The SFAM model was parametrized on PBE-D3(def2-SVP) reference data in an automated fashion, but prior to the interactive reaction study. For this, the fragmentation algorithm developed by us[31] delivers a system-focused parametrization for molecular entities of any size, with parameters being derived from optimized minimum structures of the respective fragments (for details on this procedure, we refer to Ref. 31). The total serial computing time for DFT reference data generation amounted to 3856.18 h. We note that the parametrization task is embarrassingly parallel in nature. Resorting to semiempirical methods, however, facilitates a fast parametrization even on a single desktop computer: The overall serial computing times for the SFAM parametrization then amounts to 3.58 h for a DFTB3 reference, to 4.68 h for PM6, and to 9.63 h for GFN2-xTB. A detailed discussion of the SFAM parameters derived from different reference electronic structure models can be found in the Supporting Information.

The QM region was selected based on the stochastic fragmentation algorithm described in Ref. 33 with an allowed QM region size of 15 to 50 atoms around the oxygen atom of the C-terminus of the peptide chain that is bound to the acidic proton. The reference systems to evaluate the forces of the constructed QM regions were ten systems with a size between 100 and 150 atoms. In QM/SFAM, the default charge assigned to the quantum region is 0, and the spin multiplicity is 1. However, the operator might provide an atomic information file assigning specific charges and multiplicities to specific atoms. This information will be read and automatically translated to all QM region candidates. We compared optimal QM regions proposed by our algorithm for the different semiempirical methods and the DFT reference, which all yielded the exact same optimal QM region consisting of 45 atoms, two of which are the hydrogen atoms added for valence saturation. Generally, also other selection schemes[104–109] or manual selections can be chosen, but these were not considered here. Due to the highest similarity with the DFT reference of the obtained SFAM parameters, we eventually selected DFTB3 as the ultra-fast SEQM model.

## QM/MM model for HKUST-1

The molecular structure for HKUST-1 was extracted from the Crystallography Open Database[110]. To minimize the computational cost associated with SFAM parametrization, it is preferable to carry out reference calculations only for a minimal structural motif that covers all atom types. Due to transferable atom types in the SFAM model, the generated parameters for the MOF subunit can then be transferred to a larger model of the system. The only limiting factor for structure extensions would be the incorporation of new atom types, which could be alleviated by carrying out further reference calculations, allowing an on-the-fly reparametrization[31]. This was not necessary here because of the repeating motif in the periodic structure. The HKUST-1 building unit consists of the bimetallic node and the coordinated btc-linkers (see inlay in Fig. 5). We saturated the outer carboxylate groups with hydrogen atoms to account for metal deficiencies at the boundary and ensured charge neutrality in the MM region. Hydrogen atoms at the carboxylate groups were placed with OPENBABEL[111]. Incorrectly placed hydrogen atoms at the transition metal sites were removed and the symmetrically placed hydrogen atoms on neighboring carboxylate groups were changed to be asymmetrical to improve on hydrogen bonding. Note, however, that the initial spatial placement was not crucial as these atoms were made subject to structure optimization afterward.

We selected GFN2-XTB as an electronic structure model for the QM region, which supports an interactive setting. The SFAM parametrization for the building unit and the substrates was carried out within 13 min with the GFN2-xTB reference on a desktop computer.

## Reporting summary

Further information on research design is available in the Nature Portfolio Reporting Summary linked to this article.

# Data availability

The data on the two reactions generated in this study have been deposited in the Zenodo database under accession code 10697553[112] alongside with the exploration protocols, and the Apptainer container of PUFFIN[113] that carried out the calculations for the exploration in the core system of insulin. Source data are provided with this paper.

# Code availability

The underlying SCINE software stack as well as the HERON graphical user interface are freely available and open-source[79]. A description on how to install a pre-release version of CHEMOTON, SWOOSE, and

HERON is given alongside the data archive on Zenodo[112]. In addition to the publicly available release, we note that HERON and CHEMOTON have been included into the AutoRXN workflow[59] on Microsoft Azure and Azure Quantum Elements[114,115].

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

## Acknowledgements

This publication was created as part of NCCR Catalysis, a National Centre of Competence in Research funded by the Swiss National Science Foundation (grant number 180544). M.S. gratefully acknowledges a Swiss Government Excellence Scholarship for Foreign Scholars and Artists (2020.0047).

## Author contributions

The project was conceived by all authors. K.C. and M.S. wrote the software and carried out the calculations. All authors analyzed the results and prepared the manuscript. M.S. and M.R. acquired funding. M.R. acquired the computing resources and supervised the project. M.R. is the corresponding author.

## Competing interests

The authors declare no competing interests.
