## [Peer Review File · Nature Communications]

Nanoscale chemical reaction exploration with a quantum magnifying glassREVIEWER COMMENTS

Reviewer #1 (Remarks to the Author):

This paper presents a workflow developed by the authors for reaction exploration using the QM/MM hybrid method. This workflow combines the authors' automated QM region identification algorithm, automated reaction exploration algorithm for QM regions, and interactive real-time reaction exploration model. This paper demonstrates the performance of the workflow by applying it to insulin esterification and catalytic hydrogenation by MOFs. In particular, the combination of the interactive real-time reaction exploration model and the hybrid QM/MM computational method is interesting and may be an important contribution to facilitate the theoretical analysis of complex systems. Overall, the paper is well organized and easy to understand and will appeal to a wide range of readers.

On the other hand, there are several previous reports on the combination of the automated reaction exploration algorithm and the hybrid QM/MM calculation method, but references to these are lacking. In addition, there is insufficient discussion of the consistency with experimental data in the application examples. Therefore, I would like to offer a major revision to the authors.

1. All of the barriers shown in Figure 3 are too high for biological reactions. In general, enzymatic reactions proceed smoothly at body temperature, so the barriers are often less than about 20 kcal/mol. In Figure 3, very high barriers of about 30 kcal/mol or more are obtained for all calculation methods. Careful comparisons should be made for each of the reactions considered, with reference to kinetic data (if available) and experimental conditions. If a barrier is obtained that is not realistic given the biological conditions, it is recommended that the cause of the difference be investigated.

2. In the paragraph immediately before Figure 3, it is necessary to specify which energy difference in Figure 3 corresponds to EA = 49.2 kcal/mol or EA = 31.9 kcal/mol. EA = 49.2 kcal/mol seems to be the difference between the lowest energy of the reactant in the core model IRC and the lowest energy of the TS of the core model IRC, but the energy relationship corresponding to EA = 31.9 kcal/mol cannot be found from Figure 3 (is EA = 41.9 kcal/mol wrong?). Also, I cannot find in the figure the gap EA = 29.3 kcal/mol discussed in the paragraph below Figure 3. For clarity, these numbers should be shown in the figure.

3. In this paper, the semi-empirical method is used as QM during the search and the DFT is used as QM in the last single point calculation. Both are referred to as QM, which causes confusion. Therefore, it is recommended to refer to the semi-empirical method as SEQM or the like for clarity.

4. The statement "the calculations of the stationary points of the QM/MM reaction path of the esterification of insulin and of the propylene hydrogenation in HKUST-1 were carried out with the PBE and the Minnesota M06-L exchange-correlation density functional, respectively" in the Computational methodology section is not clear enough. It needs to be clearly stated whether this is a single point calculation or a structural optimization calculation.

5. Automatic reaction exploration using the hybrid QM/MM method has also been done in [<https://doi.org/10.1021/acs.jctc.8b00799>], [<https://doi.org/10.1021/ct300633e>], and [<https://doi.org/10.1021/ct9003383>].

6. It should be mentioned that the existing automated reaction exploration methods combining QM/MM methods use the effective Hessian as a more accurate treatment than the QM block Hessian [<https://doi.org/10.1021/ct9003383>].

Reviewer #2 (Remarks to the Author):

In the manuscript by Reiher and coworkers, the authors present advances in their Scine software suite. In particular, they combine their existing tools for exploring chemical reaction networks with the automatic construction of QM/MM models. The possibilities that are provided by this methodology are demonstrated for two examples.

While the authors' software is certainly a powerful tool that can be of great use for the community (and which is available fully open source), I have some reservations concerning the publication of this manuscript in Nature Communications. Overall, the manuscript reads more like an advertisement for a software package than like a scientific paper. It mostly describes methods that are possible with the authors' software, but does not systematically validate these methods or at least compares them to existing approaches.

More specifically:

1) For the insulin example, the authors employ a workflow that consist is determining a core model, exploring possible reaction pathways for this core model, and then extending the small core model back to the large QM/MM model. However, it is never verified that the reaction paths found for the small model are actually the same as the ones that would be found for a larger model. (I understand that there are limitations in terms of computation time, but I think one can at least explore this for some example).

2) The insulin example in a pretty simple, one-step reaction. I dont understand which 850 reaction trials need to be explored here. Do these only correspond to different conformations for the reaction complex? Or are there really alternative mechanisms to explore here?

3) For the MOF example, the authors stress that their efficient exploratory strategy allows then to systematically probe different reaction pathways. This can certainly be very useful, but I think it is necessary to somehow validate this methodology. Are the reaction pathways found in the fast exploration still viable in the calculations on the larger models? And how can one be sure not to miss important pathways?

4) I do not understand what information is conveyed by Figs. 1 and 4. These look nice, but seem to have little scientific contents. In both figures, it is not clear how the different building blocks interact.

5) The possibility of exploring reactions in real time with haptic feedback is mentioned several times, but as far as I can tell, it was never used in the current work.

6) For reproducibility (and also to allow others to check and validate the authors' methodology), the coordinates of the models and the reaction pathways that are discussed should be made available.

Reviewer #3 (Remarks to the Author):

The manuscript "Quantum Magnifying Glass for Chemistry at the Nanoscale" by K.-S. Csizi, M. Steiner, and M. Reiher proposes a computational approach for transition state calculations in large systems that combines an human-guided variable quantum mechanics/molecular mechanics (QM/MM) procedure with automated transition state (TS) searches in the QM subsystem and system-specific force-field fitting in the MM subsystem. The human-in-the-loop approach is introduced as a methods of overcoming method biases in the exploration of high-dimensional potential energy surfaces (PESs).

The methodology is innovative, immediately intuitive, and of high technical quality. However, apart from the vague appeal to the "human intervention [that] can enhance process elucidation", the paper does not attempt to explain why its chosen approach improves sampling from the high-dimensional PESs and reduces bias. Two points can be made here:

* If anything, the presence of a human "operator" should increase the bias, amplifying the "operator"'s perception, which is just bias by another name. In certain situations, bias is desirable, but it is in contradiction to the stated objective of this paper.

* Additionally, one cannot stress enough how counterintuitive high-dimensional spaces are to human perception, which is trained on two- and three-dimensional objects. Unbiased exploration requires that all atomic coordinates are explored (explicitly or implicitly) in an uncorrelated fashion. Following the force, either algorithmically or by a human intervention, only generates a one-dimensional curve in the high-dimensional space.

A related issue is reproducibility. The authors write that automation is important to "ensure reproducibility and easy usability ..." but they do not make clear how this requirement is reconciled with human intervention, which will likely differ between runs. Similarly, it is unclear how the 850 trajectories of the insulin esterification reaction were obtained and if they could be reproduced in a repeat simulation.

In summary, the authors present an impressive piece of technology but cannot convincingly explain why it should be able to address the problem it is ostensibly designed to solve. I believe that this manuscript might be suitable for publication in Nature Communications after a major revision, which focuses on questions of bias, reproducibility, and transparent parameter choice. Specific questions are given below.

1. The authors should explain in the Introduction and the following sections how and why human-in-the-loop procedure enables unbiased process elucidation and address the issues raised above, including failure of human perception in high-dimensional spaces and reproducibility problems.
2. Is human steering even used in the first example (esterification reaction)? If so, how?
3. It should be made clear how the simulation parameters are chosen, for example, the 850 trajectories (reaction trials) in the esterification reaction. Are some convergence criteria used? How

are the results of the reactive trajectories (presumably not all trajectories lead to products) summarized to give a single value of activation energy and reaction energy for the esterification reaction? Can the uncertainty of the prediction be quantified? Is the uncertainty reduced with an increased number of trajectories?

4. Concerning the transition state of the esterification reaction, one would assume a two-step reaction with an high-energy tetrahedral intermediate. What does the computed activation energy of the esterification reaction correspond to: the rate-determining (highest) TS, some average or something else? What is the chemical interpretation of the averaged TS structures obtained in the calculations?

5. In the catalyzed propylene hydrogenation, how many trajectories are computed? How are the different pathways and intermediate structures in Fig. 6 distinguished? Similar to question 3, can the uncertainties related to the intermediates (both number and energies) be quantified?

Reviewer #4 (Remarks to the Author):

This article, by Csizi, Steiner, and Reiher investigates molecular substructures that lead to emergent functions. The authors introduce the concept of a "quantum magnifying glass" that automates parametrization, carries out quantum mechanical calculations, and then automates reaction explorations. They demonstrate their QM/MM reactions in proteins and MOFs, showing general applicability.

Although the article is well-written, I'd like to make some suggestions for changes prior to publication:

Major suggestions:

1. Is it actually true that 'real-time quantum chemical calculations have been limited so far to quantum systems of up to 150 atoms?' as written on page 4? The Ab Initio nanoreactor simulated a system of 228 atoms for the Urey Miller experiment. Since then, larger systems have been studied with AIMD. This should be adjusted to be more current / correct.
2. In Figure 1, the SCINE framework is introduced. However, I was not aware of this framework and I had to get to the bottom of the manuscript to know about it. Can the authors introduce SCINE earlier?
3. In figure 2, there is no X axis label. It should at least be labeled "IRC" as noted in the caption. Honestly, I am struggling to see why the IRCs have peaks at different X-axis regions. This should be clarified.
4. In the automated QM region selection procedure, the authors should indicate how they are deciding charge and spin state without human intervention. This would be particularly important for the case of the MOF that was studied, where carboxylates are negatively charged, but the cluster model is capped with hydrogen atoms. How was this automatically done? This should be detailed.

Minor suggestions:

1. Are there ways to alter the QM region selection? The current selection appears to be via atomic forces. Can charge shift analysis or other techniques be used instead?
2. Some language should be cleaned up. At some points, the QM region is referred to as the "QM region", and at other points it is called the "quantum region"
3. Although on page 8, the authors talk about the use of DFTB3, perhaps it would be useful to

understand if a machine-learned potential could be used here.

List of Changes for Manuscript ID NCOMMS-23-46509

“Quantum Magnifying Glass for Chemistry at the Nanoscale”

Katja-Sophia Csizi, Miguel Steiner and Markus Reiher

We thank the reviewers for their critical reading of the paper and for the suggestions on how to improve on it. In the following, we address the issues raised by the reviewers point by point.

Comments of Reviewer 1

This paper presents a workflow developed by the authors for reaction exploration using the QM/MM hybrid method. This workflow combines the authors’ automated QM region identification algorithm, automated reaction exploration algorithm for QM regions, and interactive real-time reaction exploration model. This paper demonstrates the performance of the workflow by applying it to insulin esterification and catalytic hydrogenation by MOFs. In particular, the combination of the interactive real-time reaction exploration model and the hybrid QM/MM computational method is interesting and may be an important contribution to facilitate the theoretical analysis of complex systems. Overall, the paper is well organized and easy to understand and will appeal to a wide range of readers.

On the other hand, there are several previous reports on the combination of the automated reaction exploration algorithm and the hybrid QM/MM calculation method, but references to these are lacking. In addition, there is insufficient discussion of the consistency with experimental data in the application examples. Therefore, I would like to offer a major revision to the authors.

We acknowledge the concerns regarding discussion of the validity of our results and will address the concerns point-by-point below (including the incorporation of lacking references).

- **1-Q1:** All of the barriers shown in Figure 3 are too high for biological reactions. In general, enzymatic reactions proceed smoothly at body temperature, so the barriers are often less than about 20 kcal/mol. In Figure 3, very high barriers of about 30 kcal/mol or more are obtained for all calculation methods. Careful comparisons should be made for each of the reactions considered, with reference to kinetic data (if available) and experimental conditions. If a barrier is obtained that is not realistic given the biological conditions, it is recommended that the cause of the difference be investigated.

- **1-A1:** *We thank the reviewer for raising the concern that the calculated barriers are too high and therefore not reflective of real-world biological reactions. We want to stress, however, that the given reaction has no biological origin, but has been constructed by us by adding a 1-propanol molecule in close proximity to the carboxylic acid group of the C-terminus of the chain (asparagine). This structural model served as the initial structure of an esterification reaction which is not catalyzed by insulin. Instead, the insulin example serves to validate a new, multi-step algorithm for analyzing chemical reactions in complex environments (our FUNNEL approach). As the algorithmic and exploratory complexity of this approach is extremely high, we have deliberately reduced the complexity of the chemical problem as much as possible. To demonstrate that the relatively high barriers do not arise from model errors, that is to show that both our QM/MM structural model and the FUNNEL strategy are reliable, we complemented our results with the following data:*

1. *QM/MM exploration: We carried out a full single-ended QM/MM exploration for the relevant esterification reaction by directly applying all search algorithms to the full structural model. Reaction trials were carried out with DFTB3/SFAM in the nanoscopic system, and single-point energies were calculated with PBE-D3(def2-SVP) on DFTB3/SFAM structures.*
2. *full QM exploration: We carried out an exploration of the esterification reaction in a full QM model. That is, insulin was calculated completely with DFTB3 without adopting a hybrid model. These data serve as a reference to validate the reliability of our QM/MM approach as well as the core model. Reaction trials were carried out with DFTB3 in the nanoscopic system, and single-point energies were calculated with PBE-D3(def2-SVP) on DFTB3 structures (again for the complete insulin structure).*

This gives us a total of 5 data sets of exploratory strategies (QM core exploration, full QM exploration, QM/MM exploration, QM/MM stationary points, QM/MM-IRC) to compare against each other in order to compensate for the lack of experimental reference data for this reaction. The activation energy of the elementary step with the lowest lying reactant energy in the full QM model is to be taken as the reference value for the given reaction. The elementary step with the lowest lying reactant energy possesses an activation energy of 45.4 kcal/mol. The same elementary step in the QM core model shows an activation energy of 42.0 kcal/mol. If the full-QM elementary step is used as an input for a QM/MM exploration, the QM/MM activation energy amounts to 45.0 kcal/mol. In the case of the QM/MM stationary points calculation, the corresponding QM/MM activation energy amounts to 51.4 kcal/mol, and to 35.7 kcal/mol for the QM/MM-IRC. The respective reaction energies amount to -7.4 kcal/mol (full QM model), -1.4 kcal/mol (QM core model), $+6.4$ kcal/mol (QM/MM exploration), $+0.4$ kcal/mol (QM/MM IRC), and -2.7 kcal/mol (QM/MM stationary points). These data are now illustrated in a new

Figure that we added to the SI:

Additional data in the box plot shown in the SI. The caption reads:

Figure 5: Box plots for the spread of relative electronic energies of the stationary points of the reaction for the core model, the 'QM/MM stationary points' of the transplanted core model into the full structure with environment relaxation, and 'QM/MM IRC' obtained from an IRC scan starting from the transition state in the full QM/MM model. The energies were calculated with PBE-D3/def2-SVP/SFAM single-point calculations on DFTB3/SFAM structures, and are given relative to the lowest reactant conformer found with each strategy. The whiskers of the boxes show the minimum and maximum values, the boxes depict the second and third quartile, and the line in the box denotes the median of the energy values. The dashed horizontal bar annotates the elementary step with the lowest reactant energy for the full QM model, and the respective elementary step in the QM/MM model for which the full-QM-model-step was used as an input. The activation energies for this elementary step in all approaches are annotated with vertical arrows.

Therefore, we conclude that full QM/MM exploration yields best quantitative agreement with the predicted activation energy, but overestimates the reaction energy to be overall endothermic. The QM/MM-IRC yields good agreement in both the activation energy and the reaction energy, which is overall a thermoneutral or slightly exothermic process, and therefore presents a reliable alternative strategy to full QM exploration, being orders of magnitude faster. We have adapted section 2.1 to these new data points, added more data to the SI, and released all raw data in a Zenodo repository.¹ For a demonstration on a chemical system with an experimental reference, we refer to the HKUST-1 data presented in section 2.2, for which we

achieved quantitative agreement (error of less than 1 kcal/mol) with experiment and existing computational studies.

- **1-Q2:** In the paragraph immediately before Figure 3, it is necessary to specify which energy difference in Figure 3 corresponds to EA = 49.2 kcal/mol or EA = 31.9 kcal/mol. EA = 49.2 kcal/mol seems to be the difference between the lowest energy of the reactant in the core model IRC and the lowest energy of the TS of the core model IRC, but the energy relationship corresponding to EA = 31.9 kcal/mol cannot be found from Figure 3 (is EA = 41.9 kcal/mol wrong?). Also, I cannot find in the figure the gap EA = 29.3 kcal/mol discussed in the paragraph below Figure 3. For clarity, these numbers should be shown in the figure.
- **1-A2:** *We selected the elementary step with the lowest reactant energy in the core model in order to discuss how our different approaches provide insights into the effect of the environment on the reaction in more detail, which serves best to validate the FUNNEL strategy. The unique identifier of the discussed elementary step is now mentioned in Table 8 of the SI along the definition of all discussed values. The EA of 49.2 kcal/mol corresponds to the activation energy of this elementary step in the core model, i.e., the energy difference between the transition state and reactant, as defined in Section 6 of the SI. The EA of 31.9 kcal/mol and 29.3 kcal/mol correspond to the activation energies for the QM/MM stationary points and QM/MM IRC approaches, respectively, with the same elementary step of the core model as an input to these approaches. In response to 1-Q1, we complemented our analyses with data from full QM, and QM/MM explorations with CHEMOTON (depicted in black and green in the revised version of Figure 3). Here, we find quantitative agreement in reaction barriers calculated with the QM core model and the full QM model, highlighting that truncating the structural model for reaction search is a valid approach in this case. For completeness, we compared QM/MM data to the elementary step with the lowest lying reactant energy of the full QM exploration (i.e. not truncating the structural model), which we added and now elaborate on in the Supporting Information. As Figure 3 condenses the data of many elementary steps into a box-plot representation, the individual reaction energies cannot be directly read from there. We therefore re-illustrated panel A of Figure 3, highlighting the discussed energies in the form of annotated arrows:*

Updated Figure 3 in the manuscript. The caption reads:

Figure 3: Reaction paths of the esterification in insulin. **A**: Box plots for the spread of relative electronic energies of the stationary points of the reaction for the core model, the 'QM/MM stationary points' of the transplanted core model into the full structure with environment relaxation, and 'QM/MM IRC' obtained from an IRC scan starting from the transition state in the full QM/MM model. The energies were calculated with PBE-D3(def2-SVP)/SFAM single-point calculations on DFTB3/SFAM structures, and are given relative to the lowest reactant conformer energy of the core model and of the single-ended QM/MM strategy for the three QM/MM strategies. The whiskers of the boxes show the minimum and maximum values, the boxes depict the second and third quartile, and the line in the box denotes the median of the energy values. The dashed horizontal bar annotates the elementary step with the lowest reactant energy in the core model, and the respective elementary step in the QM/MM model for which the core model step served as an input. The activation and reaction energies for this elementary step in all approaches are annotated with vertical arrows (for details and energy decomposition analyses, see Table 8 in the SI). **B**: Stationary points of one of the DFTB3/SFAM minimum energy reaction paths; the QM region is shown in ball-and-stick representation, whereas the MM region is presented in a stick-cartoon representation. Hydrogen atoms are omitted in the MM region for the sake of clarity.

We furthermore adapted the representation in Figure 3 such that the energies of the different QM/MM approaches can be directly compared to each other, as they are on the same potential energy surface. Previously, all energies were given relative to the lowest energy reactant conformer of each approach. Now, all QM/MM approaches share the same reference, the lowest energy reactant of the single-ended QM/MM exploration. The caption of Figure 3 now reads:

Reaction paths of the esterification in insulin. **A:** Box plots for the spread of relative electronic energies of the stationary points of the reaction for the core model, the 'QM/MM stationary points' of the transplanted core model into the full structure with environment relaxation, and 'QM/MM IRC' obtained from an IRC scan starting from the transition state in the full QM/MM model. The energies were calculated with PBE-D3(def2-SVP)/SFAM single-point calculations on DFTB3/SFAM structures, and are given relative to the lowest reactant conformer energy of the core model and of the single-ended QM/MM strategy for the three QM/MM strategies. The whiskers of the boxes show the minimum and maximum values, the boxes depict the second and third quartile, and the line in the box denotes the median of the energy values. The dashed horizontal bar annotates the elementary step with the lowest reactant energy in the core model, and the respective elementary step in the QM/MM model for which the core model step served as an input. The activation and reaction energies for this elementary step in all approaches are annotated with vertical arrows (for details and energy decomposition analyses, see Table 8 in the SI). **B:** Stationary points of one of the DFTB3/SFAM minimum energy reaction paths; the QM region is shown in ball-and-stick representation, whereas the MM region is presented in a stick-cartoon representation. Hydrogen atoms are omitted in the MM region for the sake of clarity.

We further note that we have identified a bug in our IRC approach during the revisions which prohibited the full relaxation of the IRC endpoints, which is why the data for the IRC approach have changed slightly in our revisions. However, our original conclusions are still valid. In order to reproduce both the original data and the now corrected data points, we have included both data sets explicitly labelled in the Zenodo repository,¹ containing also the Python script to create Figure 3 both in its original and now updated version.

- **1-Q3:** In this paper, the semi-empirical method is used as QM during the search and the DFT is used as QM in the last single point calculation. Both are referred to as QM, which causes confusion. Therefore, it is recommended to refer to the semi-empirical method as SEQM or the like for clarity.
- **1-A3:** *We have now updated our manuscript to be more precise. We now refer to the structural model of the protein when talking about algorithmic approaches that are independent of the underlying electronic structure as QM/MM, when we refer to individual calculations or optimizations with a semiempirical model, we refer to it as SEQM/SFAM, and when we refer to calculations with DFT, we refer to them as PBE-D3(def2-SVP)/SFAM or M06-L(def2-TZVPP)/SFAM. Furthermore, we added the following sentence to the beginning of section 2:*

Throughout this work, we rely on approximate semiempirical models for

the structure generation in the reaction exploration, which allows for more extensive sampling, but can be replaced in our modular approach by more accurate electronic structure methods. We stress that the terms 'QM' and 'MM' in 'QM/MM hybrid model' can refer in principle to any two models of which the first one can describe bond formation and dissociation reactions and the second one can describe large nanoscopic structures. In order to distinguish the actually employed models from the concept of hybrid models, we refer to the structural model as a QM/MM model, whereas we refer to the specific potential-energy models as follows: the semiempirical model combined with SFAM is denoted SEQM/SFAM and the density functional theory model as PBE-D3(def2-SVP)/SFAM (see section 4).

and complemented the caption of Figure 3 with the following sentence:

The energies were calculated with PBE-D3(def2-SVP)/SFAM single-point calculations on DFTB3/SFAM structures.

- 1-Q4: The statement "the calculations of the stationary points of the QM/MM reaction path of the esterification of insulin and of the propylene hydrogenation in HKUST-1 were carried out with the PBE and the Minnesota M06-L exchange-correlation density functional, respectively" in the Computational methodology section is not clear enough. It needs to be clearly stated whether this is a single point calculation or a structural optimization calculation.
- 1-A4: *We rewrote the corresponding sentence accordingly, which now reads:*
Single-point calculations of the stationary points of the SEQM/SFAM reaction path of the esterification of insulin and of the propylene hydrogenation in HKUST-1 were carried out with the PBE and the Minnesota M06-L² exchange-correlation density functional, respectively.
- 1-Q5: Automatic reaction exploration using the hybrid QM/MM method has also been done in [<https://doi.org/10.1021/acs.jctc.8b00799>], [<https://doi.org/10.1021/ct300633e>], and [<https://doi.org/10.1021/ct9003383>].
- 1-A5: *We thank the reviewer for pointing out these references to work from Maeda et al. and Yang et al. We rephrased the corresponding sentence in the Introduction, which now reads:*

However, these approaches are computationally demanding already for reaction mechanisms of medium-sized organic³ and inorganic⁴ compounds, and for small transition metal complexes,⁵ requiring years of computing time⁶ or specialised algorithms applicable to nanoscopic systems (as, for instance, the microiterative anharmonic downward distortion following approach introduced in Ref. 7, and the artificial-force-induced-reaction ansatz based on ONIOM 8 introduced in Ref. 9).

Furthermore, we complemented the following sentence in the Introduction with a reference to the work of Yang et al.:

Enhanced sampling approaches applied to reactions such as metadynamics,¹⁰ boxed molecular dynamics,¹¹ coordinate-driving-molecular-dynamics,¹² or the nanoreactor¹³ ...

We emphasize that existing work has leveraged QM/MM hybrid models in automated approaches, but in our present paper we introduce a new approach of targeting automated searches on a subsystem (section 2.1) and carrying out QM/MM calculations in real-time enabling interactive manipulations (section 2.2) which has not been done before.

- **1-Q6:** It should be mentioned that the existing automated reaction exploration methods combining QM/MM methods use the effective Hessian as a more accurate treatment than the QM block Hessian [<https://doi.org/10.1021/ct9003383>].
- **1-A6:** *We thank the reviewer for his reference to the effective Hessian approach. To reference the work by Maeda et al., we added the following sentence to the manuscript:*

Note that the effective Hessian approach allows for more accurate calculations of harmonic frequencies through inclusion of QM-MM interactions into the QM Hessian calculations in the form of quadratic couplings.⁷ However, this requires diagonalization of full Hessian matrices in every macroiteration step of a (transition state) optimization, including QM/QM, QM/MM, and MM/MM interaction terms, which we omit in this study for computational feasibility reasons.

Comments of Reviewer 2

In the manuscript by Reiher and coworkers, the authors present advances in their Scine software suite. In particular, they combine their existing tools for exploring chemical reaction networks with the automatic construction of QM/MM models. The possibilities that are provided by this methodology are demonstrated for two examples.

While the authors' software is certainly a powerful tool that can be of great use for the community (and which is available fully open source), I have some reservations concerning the publication of this manuscript in Nature Communications. Overall, the manuscript reads more like an advertisement for a software package than like a scientific paper. It mostly describes methods that are possible with the authors' software, but does not systematically validate these methods or at least compares them to existing approaches.

To address the reviewer's concern regarding the validation of the methods, we will reply to the comments point-by-point below:

- **2-Q1:** For the insulin example, the authors employ a workflow that consist is determining a core model, exploring possible reaction pathways for this core model, and then extending the small core model back to the large QM/MM model. However, it is never verified that the reaction paths found for the small model are actually the same as the ones that would be found for a larger model. (I understand that there are limitations in terms of computation time, but I think one can at least explore this for some example).
- **2-A1:** *We now have explored the esterification reaction applying a full QM model. As discussed also in 1-A1, the activation energy of the elementary step with the lowest lying reactant energy in this full QM model amounts to 45.4 kcal/mol. The very same elementary step in the core model amounts to 42.0 kcal/mol. The reaction energies are -1.4 (QM core) and -7.4 (full QM). The root-mean square deviations (excluding hydrogen atoms) between the QM-core and full-QM structures amount to 1.0 Å (reactant), 0.8 Å (transition state), and 1.4 Å (product). We added a Table illustrating the RMSD values of all approaches (QM/MM IRC, QM/MM stationary points, QM/MM exploration, QM-core) compared to those in the full QM model in the Supporting Information (Table 7). From these data, we can safely conclude that the **F**ocus-**U**Ntie-step in FUNNEL is a valid approach to maintain structural integrity of stationary points in the quantum region, and to deliver accurate reaction and activation energies.*
- **2-Q2:** The insulin example in a pretty simple, one-step reaction. I dont understand which 850 reaction trials need to be explored here. Do these only correspond to different conformations for the reaction complex? Or are there really alternative mechanisms to explore here?
- **2-A2:** *In fact, the number 850 is incorrect due to a mistake in a database query. The total number of executed reaction trials amounts to 510. We apologize for this error and corrected the number in the manuscript. This number can be derived from the set of reactive atoms defined in our CHEMOTON exploration and the maximum number of allowed intermolecular bond formation and dissociation reactions, plus the number of rotamers and attack points sampled. Hence, it seems too be a large number because of the unbiased exploration protocol that we apply, which does not exploit chemical insight of an operator. To detail our exploration strategy, we added a brief discussion of the applied protocol in section 4.1 of the Supporting Information.*
- **2-Q3:** For the MOF example, the authors stress that their efficient exploratory strategy allows then to systematically probe different reaction pathways. This can certainly be very useful, but I think it is necessary to somehow validate this methodology. Are the reaction pathways found in the fast exploration still viable in the calculations on the larger models? And how can one be sure not to miss important pathways?

- 2-A3: Here, there might be a misunderstanding. In section 2.2, we have **not** applied the FUNNEL strategy which explores reaction in a small subsystem, but have carried out all interactive QM/MM-explorations, QM/MM-MEP optimizations, and DFT single points on the large model that we denoted as 'extended fragment' in Figure 4 (and is also depicted in more detail in Figure 5). We acknowledge that this might not be evident from the original Figure 6 and have therefore extended its caption.

Additionally, we emphasize that there is no guarantee that all paths have been found. This is in general true for all mechanistic studies. However, the key point of our real-time approach is to deliver an increased number of viable elementary steps by facilitating the intuitive search for key intermediates. The increased number of elementary steps necessarily decreases the chance that an important pathway is missed.

This is demonstrated by the exploration of the hydrogenation mechanism. We probed (i) the two different modes of hydrogen association, (ii) the transfer to all four oxygen atoms, (iii) paths involving a 1-propyl and 2-propyl intermediate, and (iv) found new paths involving a direct hydride transfer which was missed in previous studies. Hence, we can be confident that we might have now found all important paths in our approach.

- 2-Q4: I do not understand what information is conveyed by Figs. 1 and 4. These look nice, but seem to have little scientific contents. In both figures, it is not clear how the different building blocks interact.
- 2-A4: We restructured Figure 1 to elaborate better on the sequence of actions carried out, coupled to the interplay of software modules executed in each step. The new Figure can now be read from left to right, and highlights the two complementary exploratory approaches presented in this manuscript:

Updated Figure 1 in the manuscript. The caption reads:

Figure 1: Schematic workflow of interactive reactivity exploration in quantum-classical (QM/SFAM) hybrid models. The molecule shown is chain A of the pep-

tide hormone insulin. The upper sequence of protocols (shown as blue boxes) illustrate the automated QM/MM model generation and subsequent interactive path generation and optimization protocol. The lower protocol sequence illustrate the automated generation of a QM model, followed by automated reaction search in this model, and subsequent QM/MM refinement. Relevant SCINE modules that drive the individual steps are depicted as purple gears. Employed electronic structure/ML/FF methods are depicted as white gears. All components of the modular SCINE software framework are accessible in the graphical user interface SCINE HERON, which includes, *inter alia*, a virtual environment¹⁴ for immediate visual and (optionally) haptic¹⁵⁻¹⁷ feedback denoted 'INTERACTIVE'. READUCT is a module that provides direct access to structure optimization routines based on splines and curve optimization.^{18,19} CHEMOTON²⁰ implements fully automated mechanism explorations of chemical reaction mechanisms based on first principles.

However, Figure 4 purely serves to illustrate the graphical user interface, which is a key component of the Quantum Magnifying Glass, and therefore, it is kept as is. We strongly believe that the figure should be kept to convey the message that all developments in the paper are rather easily accessible and steerable through the graphical user interface. Moreover, the figure delivers a picture of the system studied in the interactive setting (see 2-A3) and is now cross-referenced in the caption of Figure 6 in order to relate to the shown reaction network.

- 2-Q5: The possibility of exploring reactions in real time with haptic feedback is mentioned several times, but as far as I can tell, it was never used in the current work.)
- 2-A5: *We applied real-time explorations to find reaction intermediates and potential reaction paths in the hydrogenation reaction catalyzed by the MOF catalyst in section 2.2. The interactive approach coupled with a subsequent double-ended path optimization algorithm allowed us to span the reaction network depicted in Figure 6. We have added the following paragraph in section 2.2 in order to explain the real-time approach in more detail and also to clarify that this approach can be carried out without special hardware.*

Real-time calculations produce electronic energies and gradients (*i.e.*, forces) at such high frequency that they allow for instantaneous processing by an operator. Since the inspection of alpha-numerical output would require too much time for a human being to process in such a high-pace setting, it is convenient to have visual (on screen energy monitoring) and haptic (force feedback) presentations. While visual feedback requires an update rate of approximately 25 Hz for smooth real-time experience, haptic feedback (a physical force feedback upon molecule manipulation) demands a higher update rate due to the higher sensitivity of the human tactile sense. Throughout this work, we focus on real-time explorations

with a mouse-based input, allowing for the aforementioned update rate of about 25 Hz in order to achieve a real-time experience. We found that mouse-based interactions with Heron provided robust and efficient means for intuitive interactive explorations.

- 2-Q6: For reproducibility (and also to allow others to check and validate the authors' methodology), the coordinates of the models and the reaction pathways that are discussed should be made available.
- 2-A6: *We agree and have published our data in a Zenodo repository.*¹

Comments of Reviewer 3

The manuscript "Quantum Magnifying Glass for Chemistry at the Nanoscale" by K.-S. Csizi, M. Steiner, and M. Reiher proposes a computational approach for transition state calculations in large systems that combines an human-guided variable quantum mechanics/molecular mechanics (QM/MM) procedure with automated transition state (TS) searches in the QM subsystem and system-specific force-field fitting in the MM subsystem. The human-in-the-loop approach is introduced as a methods of overcoming method biases in the exploration of high-dimensional potential energy surfaces (PESs).

The methodology is innovative, immediately intuitive, and of high technical quality. However, apart from the vague appeal to the "human intervention [that] can enhance process elucidation", the paper does not attempt to explain why its chosen approach improves sampling from the high-dimensional PESs and reduces bias. Two points can be made here:

* If anything, the presence of a human "operator" should increase the bias, amplifying the "operator"'s perception, which is just bias by another name. In certain situations, bias is desirable, but it is in contradiction to the stated objective of this paper.

We acknowledge the concerns regarding the ambiguity of the meaning of bias in our manuscript and will address the concerns point-by-point below.

* Additionally, one cannot stress enough how counterintuitive high-dimensional spaces are to human perception, which is trained on two- and three-dimensional objects. Unbiased exploration requires that all atomic coordinates are explored (explicitly or implicitly) in an uncorrelated fashion. Following the force, either algorithmically or by a human intervention, only generates a one-dimensional curve in the high-dimensional space.

We agree with the reviewer and address this point below.

A related issue is reproducibility. The authors write that automation is important to "ensure producibility and easy usability ..." but they do not make clear how this requirement is reconciled with human intervention, which will likely differ between runs.

Similarly, it is unclear how the 850 trajectories of the insulin esterification reaction were obtained and if they could be reproduced in a repeat simulation.

We stress that we have developed our exploration framework with a focus on reproducibility and will address this concern below.

In summary, the authors present an impressive piece of technology but cannot convincingly explain why it should be able to address the problem it is ostensibly designed to solve. I believe that this manuscript might be suitable for publication in Nature Communications after a major revision, which focuses on questions of bias, reproducibility, and transparent parameter choice. Specific questions are given below.

- **3-Q1:** The authors should explain in the Introduction and the following sections how and why human-in-the-loop procedure enables unbiased process elucidation and address the issues raised above, including failure of human perception in high-dimensional spaces and reproducibility problems.
- **3-A1:** *Regarding reproducibility, we stress that any exploration carried out with CHEMOTON is strictly reproducible, if the exploratory protocol is given.²¹ For the insulin example, we provide the protocol as well as the results stored in a MongoDB database in the dataset uploaded to ZENODO.¹ We furthermore added a detailed discussion of the applied protocol and the resulting reaction trials (see also 2-A2) to the Supporting Information, Section 4.1.*

With respect to interactively guided reaction searches we need to clarify: These are intentionally biased to incorporate operator intuition or system prerequisites, but can be looped with READUCT or CHEMOTON. We agree with the reviewer that high-dimensional perception is difficult, but the INTERACTIVE module breaks this issue down to a 3D space. An interactively generated trajectory or structure can then be fed to READUCT, a library that implements structure and reaction path optimization routines, which provides rapid feedback on whether the initial structures are minima on a potential energy surface that are connected by a transition state.

To address the reviewer's concern about unbiased human-in-the-loop-procedures, we added the following paragraph to the Introduction:

By contrast, immediate human insight would be available if the sampling process had been presented to an operator in a way that is easy to grasp and that facilitates interactive manipulation. Here, we provide a solution that immerses an operator into a complex simulation task at the nanoscale so that human intervention can enhance chemical process elucidation. In these cases, the interaction of an operator with the exploration and its manipulation is an explicitly intended bias, which can be complemented with unbiased algorithms for confirmation of the original hypothesis.

Furthermore, we added the following paragraph to section 2.2:

As the human sense is trained on at most three-dimensional (\mathbb{R}^3) objects, it is generally counterintuitive and challenging to the human visual sense to perceive high-dimensional spaces. For instance, the $3N$ degrees of freedom of an N -atom molecule span a \mathbb{R}^{3N} space of possible reaction coordinates. As an N -atom molecule is represented in an \mathbb{R}^3 space in Heron, it becomes intuitive for the operator to grasp and manipulate structures interactively, and generate target-specific reaction paths within the \mathbb{R}^{3N} space. These paths are then handed over to automated algorithms for the optimization of QM/MM MEPs (the graphical user interface in which we implemented this setting is depicted in Figure 4).

- 3-Q2: Is human steering even used in the first example (esterification reaction)? If so, how?
- 3-A2: *We first need to clarify how the term "human steering" should be defined. If human steering refers to the real-time mouse-based structure or trajectory manipulation, then this was not applied to the esterification reaction. We intentionally selected two chemically different example systems to showcase a different workflow for each. The primary purpose of the esterification example was to introduce the FUNNEL ansatz. Therefore, the only "steering" applied here was the "Steering Wheel" algorithm²¹ incorporated into SCINE CHEMOTON, which allows an operator to guide an otherwise unbiased automated exploration. For details on this protocol, we refer to our original publication.²¹ We appreciate the reviewer's attention to detail and now ensure that the clarification regarding the use of the STEERING WHEEL in the esterification reaction example is adequately addressed in the manuscript. We therefore rephrased Section 2.1 of the manuscript accordingly, and the corresponding sentence now reads:*

Since we focus on tracing the esterification reaction pathway, we limited the searched reaction space to bimolecular reactions (excluding dimerization reactions) and restricted the reactive sites to carboxylic acid and alcohol groups. This operator-induced steering was carried out by applying the Steering Wheel algorithm.²¹

- 3-Q3: It should be made clear how the simulation parameters are chosen, for example, the 850 trajectories (reaction trials) in the esterification reaction. Are some convergence criteria used? How are the results of the reactive trajectories (presumably not all trajectories lead to products) summarized to give a single value of activation energy and reaction energy for the esterification reaction? Can the uncertainty of the prediction be quantified? Is the uncertainty reduced with an increased number of trajectories?
- 3-A3: *Regarding the number of reaction trials, we refer to our response to 2-Q2, where we elaborate on the origin of this rather high number of reaction trials for a simple one-step reaction.*

To ensure reproducibility, we added the steering protocol to the Zenodo repository that condenses the carried out exploration and all convergence criteria into a single file (given the identical version of CHEMOTON, which we also uploaded). We furthermore added a Table summarizing all applied convergence criteria to the Supporting Information (section 6.2 of the SI). In the manuscript, we do not average activation energies over different elementary steps. We therefore discuss only one selected step in greater detail, but condense the information into the box plot representation. For clarity, we added Tables including reactant, product, and transition state energies, as well as activation energies and reaction energies for each elementary step for all tested approaches (QM core exploration QM/MM exploration, QM full exploration, QM/MM stationary points, QM/MM IRC) to the Supporting Information.

With regard to uncertainty quantification we added the following paragraph to the manuscript

We emphasize that a major strength of our approach is the operator control of the depth of the exploration, which can be extended further to any degree desired, reducing uncertainties in the selected model and calculated properties. Whereas epistemic uncertainties can never be fully eliminated (see our discussion in Ref. 22), our framework allows for a flexible exchange of the underlying QM and MM model, and therefore refinement of a fast, but approximate exploration with more accurate models. Hence, the uncertainty associated with the QM/MM model choice can be decreased due to the modular character of Scine providing a range of methods. The increased requirement of resources can be reduced by applying more accurate models only on few elementary steps that are selected by a Gaussian Process model.²³

The aleatoric uncertainty, in contrast, may be reduced by sampling more trajectories. This can be achieved directly in the FUNNEL algorithm due to the initial elementary step search with Chemoton which provides multiple parameters (such as the number of sampled conformers of each reactant, number of rotamers in bimolecular reactive complexes, and the number of attack points) that can be adjusted in order to explore more trajectories. With an increased number of trajectories, more conformational degrees of freedom of the stationary points are scanned. Therefore, the likelihood increases that path associated with the lowest overall reaction energy and/or barrier was covered, and hence the uncertainties of both reaction and activation energies are reduced.

- 3-Q4: Concerning the transition state of the esterification reaction, one would assume a two-step reaction with an high-energy tetrahedral intermediate. What does the computed activation energy of the esterification reaction correspond to: the rate-determining (highest) TS, some average or something else? What is the chemical interpretation of the averaged TS structures obtained in the calculations?

- **3-A4:** *We emphasize that an esterification can proceed via multiple reaction paths, (i) a one-step mechanism where the ester bond is formed and the water simultaneously dissociates through a single transition state, (ii) the two-step mechanism mentioned by the reviewer proceeding via two transition states with a high-energy tetrahedral intermediate, and (iii) a multi-step acid or base catalyzed mechanism proceeding via charged intermediates. Because we wanted to highlight the general concept of our FUNNEL algorithm, we specifically selected a chemical reaction proceeding via a single transition state. This allowed us to explain the individual steps of the algorithm without discussing several chemical reaction steps at the same time. The elementary step discussed in the manuscript therefore corresponds to mechanism (i), which does not require any averaging of activation energies or consideration of rate-determining states. However, the very same exploration with CHEMOTON in the core model could also locate the first step of the two-step mechanism yielding the tetrahedral intermediate. We extended the exploration further and found mechanism (ii) in the core model as well, but its activation energies were similar to the one-step mechanism, which is why we added the description, exploration, and QM/MM refinement of the two-step mechanism to the Supporting Information, but believe that the main text should focus on the one-step mechanism and the explanation of the FUNNEL algorithm. Since we studied the reaction in gas phase without explicit solvent molecules, mechanism (iii) is not viable. To incorporate explicit solvation into reaction exploration is possible in principle in our framework,^{24,25} but was beyond the scope of this work.*

We now clarify the discussed mechanism in the manuscript with:

Our exploration identified 17 elementary steps of the single-step esterification given in Figure 2 A/B, among 103 elementary steps found in total. These were grouped into 18 reactions (for a detailed description, see the Supporting Information). One of the 18 reactions corresponds to the first step of a two-step esterification mechanism, yielding a tetrahedral intermediate. We extended the exploration further to complete the two-step mechanism. Given that its activation energies were similar to the one-step mechanism, we focus on the discussion of the latter. For details on the two-step mechanism, see section 4.2 of the Supporting Information.

- **3-Q5:** In the catalyzed propylene hydrogenation, how many trajectories are computed? How are the different pathways and intermediate structures in Fig. 6 distinguished? Similar to question 3, can the uncertainties related to the intermediates (both number and energies) be quantified?
- **3-A5:** *Due to the fast-paced interactive nature of the exploration process, we discarded most of the unsuccessful optimization attempts and did not record the exact number of reaction trials. A procedure for comprehensive tracking and storage of all calculations carried out during interactive explorations (including unsuccessful at-*

tempts), similar to the one presented in section 2.1, is currently under development and will be made available in a coming software release. For the given interactive exploration, we estimate that the double-ended optimization algorithm yielded a transition state in approximately 75 % of the cases. However, the number of successful reaction trials exceeded the 20 different elementary steps depicted in Figure 6. Some reaction trials led to intermediates that were not related to the hydrogenation pathway, exhibited very large activation energies, or were conformational variants of the 20 steps presented. The proposed pathways were not all documented in the literature, necessitating a trial-and-error approach based on chemical intuition. However, a single reaction trial can be completed in a few minutes on a standard desktop computer, resulting in an overall moderate computational and human effort. The values presented in Figure 6 correspond to the energetically lowest elementary step discovered for the given reaction.

Distinguishability and sorting of chemical structures is generally provided in the SCINE software framework by the library MOLASSEMBLER.²⁶ It constructs a graph representation from a three-dimensional structure and assigned chemical bonds (either based on the electronic structure such as Mayer bond orders or distance thresholds). The graph is additionally annotated at each node with the best-fitting coordination environment label (such as tetrahedral or square planar). Structures are then sorted into identical chemical compounds based on graph isomorphism. This was the underlying methodology to sort the 17 elementary steps of the one-step esterification of insulin into a single reaction as all reactants and products belong to the same chemical compounds. The analysis of intermediates along the hydrogenation pathways catalyzed by the MOF is exacerbated by two additional factors, (i) the structures are a supersystem held together by non-covalent bonds, which are not represented in MOLASSEMBLER, and (ii) the transition metal-oxygen bonds are weak with low Mayer bond orders and large bond lengths, which can introduce artificial bond formation or dissociation reactions by slight shifts around the thresholds that determine a bond. Bond order analyses are a general challenge for transition-metal complexes. Therefore, we did not solely rely on the MOLASSEMBLER classification, but additionally inspected the structures for structural similarity with quaternion fits and energetic similarity ensuring that subsequent steps in a pathway were connected by the same conformer. This increased the effort to sort the results, but (i) was deemed necessary as we noticed disconnections in the pathways in the Supporting Information in work by others²⁷ and (ii) closer manual inspections were possible given that the exploration was carried out in an interactive setting anyway.

Related to the question of the reviewer about uncertainty quantification, we refer to our answer 3-A3, but note that the interactive setting does not provide the same straightforward sampling capabilities as the FUNNEL algorithm, because the automation of this approach relies on initial human input, hence, any extension of the gained results requires additional human effort. An alternative approach would be the transfer of the reaction network obtained by interactive studies to an auto-

mated CHEMOTON exploration launched from this initial network. However, this is beyond the scope for the present work due to the earlier mentioned distinguishability challenges for this specific supersystem.

Comments of Reviewer 4

This article, by Csizi, Steiner, and Reiher investigates molecular substructures that lead to emergent functions. The authors introduce the concept of a “quantum magnifying glass” that automates parametrization, carries out quantum mechanical calculations, and then automates reaction explorations. They demonstrate their QM/MM reactions in proteins and MOFs, showing general applicability. Although the article is well-written, I’d like to make some suggestions for changes prior to publication:

Major suggestions:

- **4-Q1:** Is it actually true that ‘real-time quantum chemical calculations have been limited so far to quantum systems of up to 150 atoms?’ as written on page 4? The Ab Initio nanoreactor simulated a system of 228 atoms for the Urey Miller experiment. Since then, larger systems have been studied with AIMD. This should be adjusted to be more current / correct.
- **4-A1:** *This is a misunderstanding. ‘Real-time’ in our context does not refer to a dynamics simulation in real (compared to imaginary) time. We use this term for immediate experience and provision of results of a quantum mechanical calculation as defined in the papers by our group cited in the paper. In this respect, our statement is true. A 228-atom nanoreactor trajectory is not at all obtained in real time in the sense in which this phrase is used in our paper. However, we understand that this can be mixed up with time evolution in molecular dynamics simulations. Therefore, we have clarified this by introducing the following sentence:*

However, real-time quantum chemical calculations have been limited so far to molecular systems of up to about 150 atoms, because these computations must finish at very high frequency in order to be experienced by an operator in real time (note that this term must not be mistaken for how it is used in real (compared to imaginary) time molecular dynamics simulations). The definition of ‘real time’ in our context of interactive quantum mechanics depends on how we perceive the results as human beings: it requires an update rate for quantum chemical results of about 20 Hz for visual feedback and about 1 kHz for haptic feedback (since our tactile sense is more sensitive than the human eye) in order to not achieve real-time feedback for a smooth experience in the graphical user interface.

- 4-Q2: In Figure 1, the SCINE framework is introduced. However, I was not aware of this framework and I had to get to the bottom of the manuscript to know about it. Can the authors introduce SCINE earlier?

- 4-A2: *We complemented the introduction with a reference to the SCINE software framework. The corresponding sentence now reads:*

Key strengths of our ansatz are (i) ... and (ii) overall algorithmic flexibility, which ensures seamless mix-and-match of structural models, interaction models, and optimization algorithms, due to its incorporation into the open-source Scine software project for chemical reaction exploration²⁸ (see section 4.1 below for details).

- 4-Q3: In figure 2, there is no X axis label. It should at least be labeled "IRC" as noted in the caption. Honestly, I am struggling to see why the IRCs have peaks at different X-axis regions. This should be clarified.
- 4-A3: *Figure 2 convolutes the fitted B-Splines of the 17 elementary steps of the esterification reaction found in the exploration. As each of the elementary steps has different reactive sites with a different reaction coordinate sampled, their respective intrinsic reaction coordinates are different and therefore the plot provides no horizontal interpretability, but only in terms of the relative energies. We labeled the X-axis with "Reaction coordinates", and clarified the lack of horizontal interpretability of the data in the caption.*
- 4-Q4: In the automated QM region selection procedure, the authors should indicate how they are deciding charge and spin state without human intervention. This would be particularly important for the case of the MOF that was studied, where carboxylates are negatively charged, but the cluster model is capped with hydrogen atoms. How was this automatically done? This should be detailed.
- 4-A4: *In the QM/MM model generation step, the operator is asked to provide a molecular input structure, and (optionally) a file of atom-centered charges and corresponding spin multiplicities. For instance, if a deprotonated carboxylate group is present, one of the oxygen atoms can be assigned a charge of -1 (assignment by atom index in the full structures). For basic organic chemistry functional groups, this atomic information file can be generated automatically by applying our ASAP algorithm.²⁹ However, it can be adjusted and extended by the operator at any time. Upon selection of the quantum region, the default net charge is set to 0, and the spin multiplicity is set to 1. If any atom in the quantum region was assigned a charge or multiplicity in the atomic information file before, the total charge and multiplicity of the quantum region is adjusted accordingly. Saturation with hydrogen atoms in the quantum region will only be carried out if a covalent bond was broken. For instance, if the quantum region is capped at an O-C single bond, the oxygen atom will be saturated with a hydrogen atom. The exact positioning of these hydrogen atoms is not crucial as they will be made subject to structure optimization afterwards. For*

the MOF example, we saturated the carboxylate groups in the initial structural model with hydrogen atoms to account for metal deficiencies at the boundary and ensure overall charge neutrality in the MM region (which would otherwise adopt a charge of -18). Before starting the interactive exploration with HKUST, we compared both a singlet and triplet multiplicity for the quantum region, and selected the SEQM ground state for all calculations. We complemented the methods section of the manuscript with references to the hydrogen atom placement, and charge/multiplicity assignment:

Hydrogen atoms at the carboxylate groups were placed with OpenBabel.³⁰ Incorrectly placed hydrogen atoms at the transition metal sites were removed and the symmetrically placed hydrogen atoms on neighboring carboxylate groups were changed to be asymmetrical to improve on hydrogen bonding. Note, however, that the initial spatial placement was not crucial as these atoms were made subject to structure optimization afterwards.

and

In QM/SFAM, the default charge assigned to the quantum region is 0, and the spin multiplicity is 1. However, the operator might provide an atomic information file assigning specific charges and multiplicities to specific atoms. This information will be read and automatically translated to all QM region candidates.

The spin multiplicity was discussed in section 4.2:

The ground state spin of the HKUST-1 analog was found to be a triplet configuration with M06-L. Note that a low-spin (singlet) solution had to be adopted for the approximate GFN2-xTB tight-binding model, which is best practice for this model as it does not contain a spin-dependent energy contribution.³¹

Minor suggestions:

- 4-Q5: Are there ways to alter the QM region selection? The current selection appears to be via atomic forces. Can charge shift analysis or other techniques be used instead?
- 4-A5: *Currently, our QM/SFAM framework exploits only forces as physical quantity to select the quantum region. This is reasonable from a structural point of view (and delivers reasonable energies as demonstrated in Ref.³²). At the same time, it is computationally efficient as forces are local quantities evaluated as partial first-order derivatives at a given reference structure for each atomic nucleus. Nevertheless, it could be evaluated whether other properties should be added as complementary model selection criteria. Of course, the CSA and FSA methods introduced by Kulik et al.*

could serve as such, but would – at least in the context of this work – mitigate the full automation of our workflow, which builds on a tight interplay of SCINE software modules, in which CSA and FSA are currently not implemented. However, we appreciate feature requests on our GitHub repository if there is demand for such an extension of *Scine Swoose*.

- 4-Q6: Some language should be cleaned up. At some points, the QM region is referred to as the “QM region”, and at other points it is called the “quantum region”
- 4-A6: *To unify the terminology, we replaced “quantum region” with “QM region” throughout the manuscript and the Supporting Information.*
- 4-Q7: Although on page 8, the authors talk about the use of DFTB3, perhaps it would be useful to understand if a machine-learned potential could be used here.
- 4-A7: *We agree with the reviewer that the application of machine-learned potentials (MLPs) instead of semiempirical methods can be beneficial to accelerate QM/MM calculations, especially when the MLP is trained to represent an ab initio potential energy surface. MLPs are available in SCINE as part of our work on lifelong MLPs.³³ We have already discussed different parametrization strategies in the introduction. Additionally, we now complemented the paragraph on page 8 with the following sentence:*

In principle, recent developments of MLPs can replace expensive *ab initio* QM calculations, either by learning the difference between the results of an accurate calculation and a faster but less accurate method (denoted Δ -ML^{34,35}) or by learning the *ab initio* potential energy surface directly.³⁶ We have contributed to both approaches with ultra-fast semiempirical calculations³⁷ and a lifelong MLP which can be trained continuously on all chemical elements.³³ For both approaches, the training can be focused either on a specific system or designed in such a way that the model can generalize across multiple different systems. Hence, they are not restricted to specific chemical elements and retain high accuracy. Therefore, both approaches are suitable for replacing approximate tight-binding methods in the future.

References

- [1] Csizi, K.-S.; Steiner, M.; Reiher, M. Data set for the journal article “Quantum Magnifying Glass for Chemistry at the Nanoscale”. 2024; <https://doi.org/10.5281/zenodo.10697553>.

- [2] Zhao, Y.; Truhlar, D. G. A New Local Density Functional for Main-Group Thermochemistry, Transition Metal Bonding, Thermochemical Kinetics, and Noncovalent Interactions. *J. Chem. Phys.* **2006**, *125*, 194101.
- [3] Bensberg, M.; Reiher, M. Concentration-Flux-Steered Mechanism Exploration with an Organocatalysis Application. *Isr. J. Chem.* **2023**, *63*, e202200123.
- [4] Türtscher, P. L.; Reiher, M. Pathfinder-Navigating and Analyzing Chemical Reaction Networks with an Efficient Graph-Based Approach. *J. Chem. Inf. Model.* **2023**, *63*, 147–160.
- [5] Unsleber, J. P.; Liu, H.; Talirz, L.; Weymuth, T.; Mörchen, M.; Grofe, A.; Wecker, D.; Stein, C. J.; Panyala, A.; Peng, B.; Kowalski, K.; Troyer, M.; Reiher, M. High-Throughput Ab Initio Reaction Mechanism Exploration in the Cloud with Automated Multi-Reference Validation. *J. Chem. Phys.* **2023**, *158*, 084803.
- [6] Steiner, M.; Reiher, M. Autonomous Reaction Network Exploration in Homogeneous and Heterogeneous Catalysis. *Top. Catal.* **2022**, *65*, 6–39.
- [7] Maeda, S.; Ohno, K.; Morokuma, K. An Automated and Systematic Transition Structure Explorer in Large Flexible Molecular Systems Based on Combined Global Reaction Route Mapping and Microiteration Methods. *J. Chem. Theory Comput.* **2009**, *5*, 2734–2743.
- [8] Chung, L. W.; Sameera, W. M. C.; Ramozzi, R.; Page, A. J.; Hatanaka, M.; Petrova, G. P.; Harris, T. V.; Li, X.; Ke, Z.; Liu, F.; Li, H.-B.; Ding, L.; Morokuma, K. The ONIOM Method and Its Applications. *Chem. Rev.* **2015**, *115*, 5678–5796.
- [9] Maeda, S.; Abe, E.; Hatanaka, M.; Taketsugu, T.; Morokuma, K. Exploring Potential Energy Surfaces of Large Systems with Artificial Force Induced Reaction Method in Combination with ONIOM and Microiteration. *J. Chem. Theory Comput.* **2012**, *8*, 5058–5063.
- [10] Raucci, U.; Rizzi, V.; Parrinello, M. Discover, Sample, and Refine: Exploring Chemistry with Enhanced Sampling Techniques. *J. Phys. Chem. Lett.* **2022**, *13*, 1424–1430.
- [11] Jara-Toro, R. A.; Pino, G. A.; Glowacki, D. R.; Shannon, R. J.; Martínez-Núñez, E. Enhancing Automated Reaction Discovery with Boxed Molecular Dynamics in Energy Space. *ChemSystemsChem* **2020**, *2*, e1900024.
- [12] Yang, M.; Yang, L.; Wang, G.; Zhou, Y.; Xie, D.; Li, S. Combined Molecular Dynamics and Coordinate Driving Method for Automatic Reaction Pathway Search of Reactions in Solution. *J. Chem. Theory Comput.* **2018**, *14*, 5787–5796.
- [13] Wang, L.-P.; Titov, A.; McGibbon, R.; Liu, F.; Pande, V. S.; Martínez, T. J. Discovering Chemistry with an Ab Initio Nanoreactor. *Nat. Chem.* **2014**, *6*, 1044.

- [14] Haag, M. P.; Reiher, M. Studying chemical reactivity in a virtual environment. *Faraday Discuss.* **2014**, *169*, 89–118.
- [15] Marti, K. H.; Reiher, M. Haptic Quantum Chemistry. *J. Comput. Chem.* **2009**, *30*, 2010–2020.
- [16] Haag, M. P.; Marti, K. H.; Reiher, M. Generation of Potential Energy Surfaces in High Dimensions and Their Haptic Exploration. *ChemPhysChem* **2011**, *12*, 3204–3213.
- [17] Vaucher, A. C.; Haag, M. P.; Reiher, M. Real-Time Feedback from Iterative Electronic Structure Calculations. *J. Comput. Chem.* **2016**, *37*, 805–812.
- [18] Vaucher, A. C.; Reiher, M. Minimum Energy Paths and Transition States by Curve Optimization. *J. Chem. Theory Comput.* **2018**, *14*, 3091–3099.
- [19] Heuer, M. A.; Vaucher, A. C.; Haag, M. P.; Reiher, M. Integrated Reaction Path Processing from Sampled Structure Sequences. *J. Chem. Theory Comput.* **2018**, *14*, 2052–2062.
- [20] Unsleber, J. P.; Grimmel, S. A.; Reiher, M. Chemoton 2.0: Autonomous Exploration of Chemical Reaction Networks. *J. Chem. Theory Comput.* **2022**, *18*, 5393–5409.
- [21] Steiner, M.; Reiher, M. Navigating chemical reaction space with a steering wheel. 2023; <https://arxiv.org/abs/2308.16499>.
- [22] Csizi, K.-S.; Reiher, M. Universal QM/MM Approaches for General Nanoscale Applications. *WIREs Comput. Mol. Sci.* **2023**, *13*, e1656.
- [23] Simm, G. N.; Reiher, M. Error-Controlled Exploration of Chemical Reaction Networks with Gaussian Processes. *J. Chem. Theory Comput.* **2018**, *14*, 5238–5248.
- [24] Simm, G. N.; Türtscher, P. L.; Reiher, M. Systematic Microsolvation Approach with a Cluster-Continuum Scheme and Conformational Sampling. *J. Comput. Chem.* **2020**, *41*, 1144–1155.
- [25] Bensberg, M.; Türtscher, P. L.; Unsleber, J. P.; Reiher, M.; Neugebauer, J. Solvation Free Energies in Subsystem Density Functional Theory. *arXiv:2108.11228 [cond-mat, physics:physics]* **2021**,
- [26] Sobez, J.-G.; Reiher, M. Molassembler: Molecular Graph Construction, Modification, and Conformer Generation for Inorganic and Organic Molecules. *J. Chem. Inf. Model.* **2020**, *60*, 3884–3900.
- [27] Shakya, D. M.; Ejegbavwo, O. A.; Rajeshkumar, T.; Senanayake, S. D.; Brandt, A. J.; Farzandh, S.; Acharya, N.; Ebrahim, A. M.; Frenkel, A. I.; Rui, N.; Tate, G. L.; Monnier, J. R.; Vogiatzis, K. D.; Shustova, N. B.; Chen, D. A. Selective Catalytic Chemistry at Rhodium(II) Nodes in Bimetallic Metal–Organic Frameworks. *Angew. Chem. Int. Ed.* **2019**, *58*, 16533–16537.

- [28] Software for Chemical Interaction and Networks (SCINE). <https://scine.ethz.ch/>, accessed in February 2024.
- [29] Csizi, K.-S.; Reiher, M. Automated preparation of nanoscopic structures: Graph-based sequence analysis, mismatch detection, and pH-consistent protonation with uncertainty estimates. *J. Comp. Chem.* **2023**, *Early View*.
- [30] O’Boyle, N. M.; Banck, M.; James, C. A.; Morley, C.; Vandermeersch, T.; Hutchison, G. R. Open Babel: An Open Chemical Toolbox. *J. Cheminformatics* **2011**, *3*, 33.
- [31] Bannwarth, C.; Caldeweyher, E.; Ehlert, S.; Hansen, A.; Pracht, P.; Seibert, J.; Spicher, S.; Grimme, S. Extended Tight-Binding Quantum Chemistry Methods. *WIREs Comput. Mol. Sci.* **2021**, *11*, e1493.
- [32] Brunken, C.; Reiher, M. Automated Construction of Quantum–Classical Hybrid Models. *J. Chem. Theory Comput.* **2021**, *17*, 3797–3813.
- [33] Eckhoff, M.; Reiher, M. Lifelong Machine Learning Potentials. *J. Chem. Theory Comput.* **2023**, *19*, 3509–3525.
- [34] Ramakrishnan, R.; Dral, P. O.; Rupp, M.; von Lilienfeld, O. A. Big Data Meets Quantum Chemistry Approximations: The Δ -Machine Learning Approach. *J. Chem. Theory Comput.* **2015**, *11*, 2087–2096.
- [35] Dral, P. O. Quantum Chemistry in the Age of Machine Learning. *J. Phys. Chem. Lett.* **2020**, *11*, 2336–2347.
- [36] Behler, J.; Csányi, G. Machine learning potentials for extended systems: a perspective. *Eur. Phys. J. B* **2021**, *94*, 1–11.
- [37] Bosia, F.; Zheng, P.; Vaucher, A.; Weymuth, T.; Dral, P. O.; Reiher, M. Ultra-Fast Semi-Empirical Quantum Chemistry for High-Throughput Computational Campaigns with Sparrow. *J. Chem. Phys.* **2023**, *158*, 054118.

REVIEWER COMMENTS

Reviewer #1 (Remarks to the Author):

The authors have adequately addressed most of my comments. I am still puzzled as to why the authors avoid application to real biological reactions with accurate experimental kinetic data. As a computational chemist with experience in modeling enzyme reactions and knowing how difficult it is to model enzyme reactions accurately, I even doubt that the real reason is that their approach is not accurate enough for direct comparison with experimental data in biological examples. I recommend that the authors add an adequate explanation for this doubt and hope that further validations will be done in the future. In any case, this is a nice work and I can now recommend the paper for publication in Nature Communications.

Reviewer #2 (Remarks to the Author):

The authors have addressed all my previous comments. With these changes (and the changes made in response to the other reviewers, who addressed similar points), I believe the manuscript is now suitable for publication.

Reviewer #3 (Remarks to the Author):

In the revised version of the manuscript "Quantum Magnifying Glass for Chemistry at the Nanoscale" by K.-S. Csizi, M. Steiner, and M. Reiher, the authors clarified the details of their approach, provided some measures of uncertainty, and added a discussion of potential biases due to human intervention or the methods choices (such as restrictions to specific reaction mechanisms). I believe that these biases are a fundamental limitation of the proposed methodology. On the other hand, the approach is of interest to exploration of reaction mechanisms in large systems and should be published. I would ask the authors to address the following minor points before publication.

1. The authors should be more explicit in the Introduction section about the heuristics still used in their "quantum magnifying glass" algorithm, specifically the choice of reactive sites and the application of heuristic reaction rules. The "magnification" is brought about in this case by the heuristic choices that the authors made ahead of time. One can see it in the fact that they observed a two-step esterification mechanism when they went looking for it.

This important detail is mentioned as a hypothetical in Sec. 2.1 (p. 7, "First, the underlying esterification mechanism is chemically well-defined and can be encoded into a limited set of rules to be applied in the exploration protocol, hence keeping the search space comparatively small.") and then in passing later in Sec. 2.1 (p. 9, "Since we focus on tracing the esterification reaction pathway, we limited the searched reaction space to bimolecular reactions (excluding dimerization reactions) and

restricted the reactive sites to carboxylic acid and alcohol groups.") before waving away the impact of these choices, simply stating "Whereas epistemic uncertainties can never be fully eliminated, ...". The authors conflate the definitions of uncertainty and bias, which are two very different effects. In fact, the heuristics chosen in this work do not increase the "epistemic uncertainty" but create a systematic bias.

I would ask the authors to explicitly include a mention of the heuristic rules in the Introduction section and show their impact on the reported results in Sec. 2.1, leading to the "discovery" of the two-step mechanism in Sec. 2.1. The reaction rules used in the propylene hydrogenation calculations (Sec. 2.2) should be mentioned in the main text or the supplementary information. I could not find them in either place.

2. The term "elementary step" should be clearly defined. It seems that the authors mean by "elementary step" an energy profile having the expected shape. If this is the definition, it is easy to confuse with the definition of elementary steps in multi-step reaction mechanisms. This should be clarified.

Reviewer #3 (Remarks on code availability):

Both code and calculation data are provided in the online archive and give the full view of the results presented in the paper. The source code is well structured and documented, allowing to read and understand the algorithm.

Reviewer #4 (Remarks to the Author):

The authors have addressed my comments sufficiently. I commend the authors for making the necessary changes.

List of Changes for Manuscript ID NCOMMS-23-46509

“Quantum Magnifying Glass for Chemistry at the Nanoscale”

Katja-Sophia Csizi, Miguel Steiner and Markus Reiher

We thank the reviewers for their critical re-reading of the paper including the recent revisions. In the following, we address the remaining issues raised by the reviewers point by point.

Comments of Reviewer 1

The authors have adequately addressed most of my comments. I am still puzzled as to why the authors avoid application to real biological reactions with accurate experimental kinetic data. As a computational chemist with experience in modeling enzyme reactions and knowing how difficult it is to model enzyme reactions accurately, I even doubt that the real reason is that their approach is not accurate enough for direct comparison with experimental data in biological examples. I recommend that the authors add an adequate explanation for this doubt and hope that further validations will be done in the future. In any case, this is a nice work and I can now recommend the paper for publication in Nature Communications.

- *The insulin example was chosen with a focus on simplicity to introduce fundamental aspects of our methodology. The transferability of the QMG, especially FUNNEL, to more complex enzymatic reactions must and will be carefully assessed in future work, especially in multi-step catalytic cycles involving transition metal ions, which generally present significant challenges for modeling enzymatic reactions. For instance, one critical aspect is the choice of the QM and MM models, which will determine the overall accuracy and the model fidelity when compared with experiment. Although the modular character of SCINE provides various options of highly accurate electronic structure models, their thorough validation for an enzymatic reaction with proper solvation is simply beyond the scope of our current study. To explain this situation in some detail, we added the following sentence to the conclusion:*

The transferability of the QMG to more complex enzymatic reactions needs to be carefully assessed. Especially the exploration of multi-step catalytic cycles involving transition metal ions can present significant challenges to the electronic structure model employed. Since ultra-fast SEQM/MM models trade speed for accuracy, more accurate electronic structure methods will be required for refinement, but at the expense of significantly increased computational resource requirements. Although

this can be done in an automated fashion in the background then, it requires an effort that is beyond the scope of the present work.

Comments of Reviewer 2

The authors have addressed all my previous comments. With these changes (and the changes made in response to the other reviewers, who addressed similar points), I believe the manuscript is now suitable for publication.

- *No further changes required.*

Comments of Reviewer 3

In the revised version of the manuscript "Quantum Magnifying Glass for Chemistry at the Nanoscale" by K.-S. Csizi, M. Steiner, and M. Reiher, the authors clarified the details of their approach, provided some measures of uncertainty, and added a discussion of potential biases due to human intervention or the methods choices (such as restrictions to specific reaction mechanisms). I believe that these biases are a fundamental limitation of the proposed methodology. On the other hand, the approach is of interest to exploration of reaction mechanisms in large systems and should be published. I would ask the authors to address the following minor points before publication.

- **3-Q1:** The authors should be more explicit in the Introduction section about the heuristics still used in their "quantum magnifying glass" algorithm, specifically the choice of reactive sites and the application of heuristic reaction rules. The "magnification" is brought about in this case by the heuristic choices that the authors made ahead of time. One can see it in the fact that they observed a two-step esterification mechanism when they went looking for it.

- **3-A1:** *We agree that both the FUNNEL approach and an interactive exploration introduce a bias, but this is a wanted bias and a feature of the algorithm. It allows one to quickly explore expected reactions or probe potentially relevant reactions efficiently. The findings can be subjected to further (semi-)automated explorations within our framework.*

An unbiased sampling of the PES of nanoscopic systems is computationally unfeasible (even enhanced sampling algorithms, which seem to be unbiased, actually introduce biases (e.g., by limiting the reactions found to a single PES or by introducing collective variables which are thought to capture the relevant reaction chemistry).

We argue that the FUNNEL approach in principle allows the operator to expand the search space continuously (limited only by computational resources) due to its

high degree of autonomy, allowing the operator to decrease the bias, if deemed necessary. However, we did not demonstrate this explicitly as it is out of the scope of the present work, but otherwise a rather straightforward extension.

For the capabilities of CHEMOTON and STEERING WHEEL available for more exhaustive explorations, we refer to Refs. 1 and 2. The interactive approach in section 2.2, by contrast, is based on chemical intuition as an intended bias, which is introduced by real-time manipulations. To discuss this, we added the following sentence to the introduction:

As the unbiased sampling of the PES of a nanoscopic is computationally unfeasible, we introduce an intended bias focused on the given problem. Within the QMG, this bias can take the form of (i) interactive manipulations of structures in real-time by an operator and (ii) *a priori* defined reaction heuristics that guide an automated exploration. The former comes with the advantage of offering instant feedback on the reactivity of a system, allowing for an adjustment and hence improvement of the exploration coverage. The latter can be advantageous because introducing reaction rules can guide an initial automated mechanism exploration that subsequently extends its scope by systematically lifting or altering these reaction rules. We note that such biases are also present in seemingly unbiased enhanced sampling techniques (consider, for instance, the choice of velocities for the nanoreactor³ or the choice of collective variables in metadynamics⁴). However, such enhanced sampling techniques restrict an exploration to one PES whereas our approach allows for an easy introduction of new reagents and reactants.

Furthermore, we realized that the applied reaction heuristics and rules might have been insufficiently explained in the previous versions of our paper. To clarify the heuristic rules applied in the FUNNEL approach, we added the following paragraph to Section 2.1 (note that we refrained from describing the heuristic rules in detail in the introduction as they only apply to the insulin example (section 2.1), but not to the H-KUST example (section 2.2); see 3-A3 below):

Note that the heuristic rules applied through the Steering Wheel were formulated as general as possible in order to minimize the truncation of the search space. Therefore, we only defined (i) reactive sites, that is atom types in functional groups that are considered reactive, and determined the sampled reaction coordinates, and (ii) a total number of bond dissociation and formation reactions in the entire reactive complex. No atom-specific changes in the connectivity were enforced. By that, all reactions between carbon and/or oxygen atoms in a carboxylate group, and oxygen and/or hydrogen atoms in a hydroxyl group were tested.

- 3-Q2: This important detail is mentioned as a hypothetical in Sec. 2.1 (p. 7, "First, the underlying esterification mechanism is chemically well-defined and can be encoded into a limited set of rules to be applied in the exploration protocol,

hence keeping the search space comparatively small.”) and then in passing later in Sec. 2.1 (p. 9, ”Since we focus on tracing the esterification reaction pathway, we limited the searched reaction space to bimolecular reactions (excluding dimerization reactions) and restricted the reactive sites to carboxylic acid and alcohol groups.”) before waving away the impact of these choices, simply stating ”Whereas epistemic uncertainties can never be fully eliminated, ...”. The authors conflate the definitions of uncertainty and bias, which are two very different effects. In fact, the heuristics chosen in this work do not increase the ”epistemic uncertainty” but create a systematic bias.

- **3-A2:** *We thank the reviewer for highlighting that our definitions of uncertainty and bias are conflated, a problem that emerged during the revision process. We complemented and rephrased the mentioned paragraph, which now reads:*

The exploratory bias arising from the choice of reaction rules is a source of aleatoric uncertainty. By continuously expanding the set of applied heuristic rules and therefore increasing the exploration breadth, the bias can be decreased systematically. Even in the case of very extensive sampling, epistemic uncertainties can never be fully eliminated (see our discussion in Ref. 5). Therefore, our framework allows for a flexible exchange of the underlying QM and MM model, and therefore refinement of a fast, but approximate exploration with more accurate models.

The impact of the choice of reactive sites and reaction rules is discussed in our response to 3-Q1 and 3-Q3.

- **3-Q3:** I would ask the authors to explicitly include a mention of the heuristic rules in the Introduction section and show their impact on the reported results in Sec. 2.1, leading to the ”discovery” of the two-step mechanism in Sec. 2.1. The reaction rules used in the propylene hydrogenation calculations (Sec. 2.2) should be mentioned in the main text or the supplementary information. I could not find them in either place.
- **3-A3:** *With regard to the H-KUST example, no predefined heuristic rules have been applied in the interactive study of the propylene hydrogenation reactions (in contrast to the FUNNEL approach). One of the interactive framework’s principles is that a reaction trial should remain continuously and dynamically adaptable, based on preceding trials or results, or simply based on chemical intuition. In contrast to FUNNEL, where an extremely large number of reactions can be probed systematically and largely autonomously, the number and type of reaction trials in the interactive setting are determined exclusively by the input of the operator, which can, however, be adapted continuously. Apart from integrating chemical intuition into the exploration process, the interactive approach enables rapid ”changes of focus” in an exploration. For example, when evaluating the hydrogenation reactions presented in this manuscript, our input to the software was mainly based on the outcome of the previous reaction trials, which allowed us to find new pathways that*

were not yet discussed in the literature. With respect to the reviewer’s concern regarding the introduction of bias, we added the following sentence to Section 2.2:

In this example, the exploration bias did not result from the automated application of predefined reactivity rules as in the FUNNEL approach, but from the interactive nature of the exploration, where only the reaction paths interactively probed by the operator were evaluated.

Additionally, we want to clarify that our previous revisions did not include any changes to the reaction heuristics in order to find the two-step mechanism. Instead, we continued the automated exploration with the same reaction rules as before, so we only changed the exploration depth as required for multi-step reaction mechanisms. This was possible due to the general formulation of our rules. We agree with the reviewer that incorrectly biased explorations can miss important elementary steps. We would like to stress that any decision made during a mechanistic study introduces a bias that can compromise the results. However, our automation delivers the advantage of studying many more elementary steps than conventional approaches make accessible while explicitly defining the biases, hence reducing the odds of missing important elementary steps. We added the following sentence in section 2.1 to clarify our procedure to explore the two-step mechanism:

One of the 18 reactions corresponds to the initial step of a two-step esterification mechanism, yielding a tetrahedral intermediate. Given that our heuristic reaction rules were defined in a general manner in terms of functional groups, we were able to extend the exploration straightaway with unaltered reaction rules allowing for unimolecular reactions of the kinetically favored products of the exploration up to that point. This extension of the exploration completed the two-step mechanism.

- 3-Q4: The term "elementary step" should be clearly defined. It seems that the authors mean by "elementary step" an energy profile having the expected shape. If this is the definition, it is easy to confuse with the definition of elementary steps in multi-step reaction mechanisms. This should be clarified.
- 3-A4: *We thank the reviewer for highlighting that the term "elementary step" is not clearly defined. We therefore added the following clarification to Section 2.1 of the manuscript*

Elementary steps connect (sets of) structures either via a single transition state (first-order saddle point on the PES) or a barrierless association or dissociation reaction. A structure is a stationary point on the PES that corresponds to a three-dimensional arrangement of nuclei with a given molecular charge and spin multiplicity. Structures that are local minima on the PES can be grouped into compounds if they share the same nuclear composition and connectivity, but differ in the three-dimensional arrangement. Accordingly, elementary steps can be

grouped into reactions, if the associated reactant and product structures belong to the same reactant and product compounds. For details on these definitions, we refer to Ref. 6.

Both code and calculation data are provided in the online archive and give the full view of the results presented in the paper. The source code is well structured and documented, allowing to read and understand the algorithm.

- *We thank the reviewer for reviewing the supplementary data and software, as well as their detailed review of the revised manuscript.*

Comments of Reviewer 4

The authors have addressed my comments sufficiently. I commend the authors for making the necessary changes.

- *No further changes required.*

References

- [1] Unsleber, J. P.; Grimmel, S. A.; Reiher, M. Chemoton 2.0: Autonomous Exploration of Chemical Reaction Networks. *J. Chem. Theory Comput.* **2022**, *18*, 5393–5409.
- [2] Steiner, M.; Reiher, M. Navigating chemical reaction space with a steering wheel. 2023; <https://arxiv.org/abs/2308.16499>.
- [3] Wang, L.-P.; Titov, A.; McGibbon, R.; Liu, F.; Pande, V. S.; Martínez, T. J. Discovering Chemistry with an Ab Initio Nanoreactor. *Nat. Chem.* **2014**, *6*, 1044.
- [4] Ray, D.; Parrinello, M. Kinetics from Metadynamics: Principles, Applications, and Outlook. *J. Chem. Theory Comput.* **2023**, *19*, 5649–5670.
- [5] Csizi, K.-S.; Reiher, M. Universal QM/MM Approaches for General Nanoscale Applications. *WIREs Comput. Mol. Sci.* **2023**, *13*, e1656.
- [6] Unsleber, J. P.; Reiher, M. The Exploration of Chemical Reaction Networks. *Annu. Rev. Phys. Chem.* **2020**, *71*, 121–142.

REVIEWERS' COMMENTS

Reviewer #1 (Remarks to the Author):

The authors answered all my questions adequately. I can recommend the paper for publication.

Reviewer #3 (Remarks to the Author):

I appreciate the authors' revisions, which have addressed my comments in detail. Thus I am pleased to recommend the revised manuscript for publication in Nature Communications.